# Recent Advancements of UF-Based Separation for Selective Enrichment of Proteins and Bioactive Peptides—A Review

**Enny Ratnaningsih [1], Reynard Reynard [2,3], Khoiruddin Khoiruddin [3], I Gede Wenten [3,4,\*] and Ramaraj Boopathy [5,\*]**

1   Biochemistry Research Division, Faculty of Mathematics and Natural Sciences, Institut Teknologi Bandung, Jl. Ganesha No. 10, Bandung 40132, Indonesia; enny@chem.itb.ac.id
2   Department of Agrotechnology and Food Sciences, Wageningen University and Research, 6708WG Wageningen, The Netherlands; reynard.reynard@wur.nl
3   Department of Chemical Engineering, Faculty of Industrial Technology, Institut Teknologi Bandung, Jl. Ganesha No. 10, Bandung 40132, Indonesia; khoiruddin@che.itb.ac.id
4   Research Center for Bioscience and Biotechnology, Institut Teknologi Bandung, Jl. Ganesha No. 10, Bandung 40132, Indonesia
5   Department of Biological Sciences, Nicholls State University, Thibodaux, LA 70301, USA
\*   Correspondence: igw@che.itb.ac.id (I.G.W.); ramaraj.boopathy@Nicholls.edu (R.B.); Tel.: +1-(985)-448-4716 (R.B.)

**Featured Application: Proteins could be beneficial in their concentrated products or isolates, of which mem-brane-based filtration methods, such as ultrafiltration (UF), encompass application in broad spectra of protein sources.**

**Abstract:** Proteins are one of the primary building blocks that have significant functional properties to be applied in food and pharmaceutical industries. Proteins could be beneficial in their concentrated products or isolates, of which membrane-based filtration methods such as ultrafiltration (UF) encompass application in broad spectra of protein sources. More importantly, selective enrichment by UF is of immense interest due to the presence of antinutrients that may dominate their perspicuous bioactivities. UF process is primarily obstructed by concentration polarization and fouling; in turn, a trade-off between productivity and selectivity emerges, especially when pure isolates are an ultimate goal. Several factors such as operating conditions and membrane equipment could leverage those pervasive contributions; therefore, UF protocols should be optimized for each unique protein mixture and mode of configuration. For instance, employing charged UF membranes or combining UF membranes with electrodialysis enables efficient separation of proteins with a similar molecular weight, which is hard to achieve by the conventional UF membrane. Meanwhile, some proposed strategies, such as utilizing ultrasonic waves, tuning operating conditions, and modifying membrane surfaces, can effectively mitigate fouling issues. A plethora of advancements in UF, from their membrane material modification to the arrangement of new configurations, contribute to the quest to actualize promising potentials of protein separation by UF, and they are reviewed in this paper.

**Keywords:** bioactivity; concentration polarization; fouling; surface modification

## 1. Introduction

Proteins are essential to maintain proper energy density of living beings, along with regulating activities of enzymes pertinent to type-2 diabetes, hypertension, and stress relief [1]. In other purposes, proteins have ample industrial relevancies as biocatalysts and food additives, i.e., emulsifying agent, flavor enhancement, etc. [2,3]. In recent times, protein hydrolysates have gained popularity since they improve digestibility by increasing protein solubility and diminishing prominent antinutrients. Furthermore, bioactive peptides are released by protein hydrolysis. In order to activate these bioactive peptides,

they should be separated from cognate proteins [4,5]; membrane filtration methods play a paramount role for their high throughput compared to other separation processes. Methods of separation of proteins using extraction, precipitation, centrifugation, and chromatography are common in practice, but hindered in several ways, such as by denaturation and proteolysis (of extraction and precipitation) [6–8], resolution challenges of centrifugation [9], and low yield of chromatography-based techniques [10]. Beneficially, membrane technology offers mild filtration conditions (low temperature and pressure) without phase change, retaining bioactivity [11]. Furthermore, these membrane filtration methods also possess interesting features such as molecular separation, high separation efficiency, lower footprint, less chemical consumption, and easy scale-up, which promote their application in various fields [12–15].

Applications of membrane filtration in protein separation and concentration are omnipresent either in bench-scale production or industries. Various membrane applications to isolate specific peptides with particular functions are given in Tables 1–3. Initially, conventional use of pressure-driven membrane-based processes is adapted to separate molecules with different size; microfiltration (MF), for example, is carried out to remove cell debris of fermentation media or hydrolysates with a molecular size of about 0.1–10 μm. Conventional ultrafiltration (UF) is also size-based and exerts low-resolution requirement owing to strict rule-of-thumb to only separate molecules with tenfold differences [16,17]. Therefore, multitudes of experimental works, which have been integrated into commercial processes, are initiated to advance UF with a combined driving force (electrical and/or concentration). These do not depend only on pressure gradients. For example, fractionation and/or purification of proteins in the range of 0.1–5 μm could be fractionated by charged UF, peptides around 1–10 μm by membrane chromatography, and that with 0.1–5 nm in size may be gathered by ultrasound (US)-assisted UF, high-performance tangential flow filtration (HPTFF), electro-UF membrane (EUF), electrodialysis with UF membrane (EDUF), and electrodialysis with bipolar membrane (EDBM) [18,19].

The advent of those novel advancements in UF is owing to the bigger challenges in managing the trade-off between productivity and selectivity. In general, interests in advancing UF in protein separation are steadily increasing, as shown in Figure 1. It is obvious that published research studies in UF for peptide separation (see the inset) are always lower than UF for protein separation due to the setbacks of UF to achieve high selectivity of small peptide fractions, regardless of their higher productivity. In terms of research focus, presented in Figure 1b, the top four burgeoning preferences are in the area of membrane modifications (26%), charged-membrane UF (15%), EDUF, (15%), and dynamic UF (15%). Membrane modifications and dynamic UF are sought to tackle the issue of fouling with the best scale-up possibilities. Meanwhile, EDUF and charged-membrane UF are thriving, which can be attributed to the fact that proteins or peptides with similar size, primarily those which have lower molecular weight (the most valuable owing to their dominant bioactivity), are the most difficult to separate by traditional processes and need cutting-edge developments.

There are several reviews related to protein separation by filtration and UF in particular, such as thorough membrane-based protein separation [19,20], UF application in food industries [21] and in general applications [22], alfalfa leaf protein recovery by filtration [23], whey protein separation and purifications by membrane processes [18,24,25], recovery of protein from fish meal wastewater by ultrafiltration [26], recovery of antihypertensive peptides by membrane-based production [27], pore blocking in ultrafiltration [28], fouling in MF and UF [29,30], effect of ultrasonic on membrane filtration [31], specific review on EDUF [32], and dynamic filtration [33]. There are no comprehensive reviews, however, covering definitive accounts of protein separation and purification by specific means of UF. In this paper, the main theme of our objective is hence to present recent advancements in UF membrane processes, from charged-membrane UF, EUF, EDUF, EDBM, ultrasonic-assisted UF, dynamic UF, and HPTFF to UF integration. Following them, the number of strategies in controlling fouling and concentration polarization are presented from the aspect of mem-

brane modifications to tailoring operation conditions: concentration, pH, ionic strength, transmembrane pressure (TMP), and temperature. Ultimately, the potentials of using UF as primary setup or complementary protocols to select the most potent bioactive peptides are described in brief in the final section.

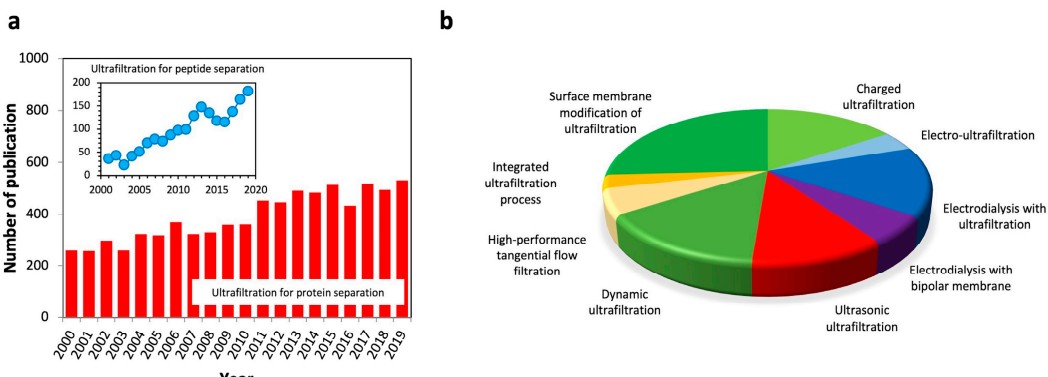

**Figure 1.** (**a**) Number of publications associated with "protein ultrafiltration" and "peptide ultrafiltration" as indexed by Scopus [TITLE-ABS-KEY (terms)]. (**b**) Research focus in ultrafiltration of protein, divided by various [KEY (terms)] of Scopus showed in the legend.

**Table 1.** Significant ultrafiltration (UF)-assisted separation process of protein hydrolysates with expressed bioactivities from plant sources.

| Source | Hydrolysis Enzyme(s) | Protocols | Bioactivity | Target Compound (s) | Remarks | Ref |
|---|---|---|---|---|---|---|
| *Grateloupia turuturu* (Macroalgae) | - | Preindustrial UF PCI-MT600 (Polyethersul-fone (PES) 30 kDa) from | -Antioxidant | R-phycoerythrin | 100% recovery rate flux: 35.1 L m$^{-2}$ h$^{-1}$ | [34] |
| *Enteromorpha clathrata* (Macroalgae) | Alcalase | QuixStand$^{TM}$ Benchtop System (10 kDa) | -ACE inhibitors | Peptide fragments: PAFG | Yield: 0.82% IC$_{50}$: 35.9 µm | [35] |
| Pigeon pea | Pepsin–pancreatin, Alcalase, or Pancreatin | Sequential UF by stirred cell Amicon 8400 (1, 3, 5, 10 kDa) | -Antioxidant (inhibitor lipid peroxidation) | Peptides < 1 kDa Peptides < 5 kDa | Yield: 36.97% (<1 kDa from pepsin–pancreatin hydrolysate), 28.82% (1–3 kDa, Alcalase), 37.27% (1–3 kDa, pancreatin) | [36] |
| Soy protein | Flavourzyme | Sequential UF (30, 10, 1, 0.3 kDa) | -Lipolysis stimulator | Peptides ~1 kDa | Intracellular triglyceride decrease (2.73 to 2.30 mol mg$^{-1}$ protein) by 400 ppm peptides | [37] |
| *Vigna unguiculata* (Cowpea) | Alcalase/ Flavourzyme/ pepsin–pancreatin | Sequential UF high performance UF cell (Model 2000, Millipore) (1, 3, 5, 10 kDa) | -ACE inhibitors -Antioxidants | Peptides < 1 kDa | ACE inhibitors: Alcalase: 24.3–123, Flavourzyme: 0.04–170.6, pancreatin: 44.7–112 µg mL$^{-1}$ (IC$_{50}$) Antioxidant: Alcalase: 303.2–1457; Flavourzyme: 357.4–10,211; Pancreatin hydrolysate: 267.1–2830.4 (Trolox equivalent) | [38] |

**Table 1.** *Cont.*

| Source | Hydrolysis Enzyme(s) | Protocols | Bioactivity | Target Compound (s) | Remarks | Ref |
|---|---|---|---|---|---|---|
| Soybean meal fermented by *Bacillus subtilis* | Thermolysin | Sequential filtration (MF: fractions of > 10 kDa, UF: 1–10, NF: 0.5–1, RO: <0.5 kDa) | Antihypertensive | Peptides < 0.5 kDa, Fragments: QC, PANV, GPANV | Endothelium-independent vasorelaxation action (work similarly with ACE-I inhibitory peptides) | [39] |
| Soy protein | Flavourzyme® or chymotrypsin | Sequential UF (Regenerated cellulose (RC) 50, 10, and 3 kDa) | Antioxidants | Peptides < 3 kDa | Peptides from both hydrolysates decreased liposome oxidation by 83.2% and 84.5%, and showed DPPH activity of 21.3 and 24.4%, respectively | [40] |
| *Phaseolus lunatus* and *Proteus vulgaris* | Alcalase | Sequential UF high-performance UF cell (Model 2000, Millipore) (1, 3, 5, 10 kDa) | ACE inhibitors | Peptides < 1 kDa | -Activity determined by higher constituents of hydrophobic amino acids such as valine and proline -IC$_{50}$ (µg mL$^{-1}$): *Phaseolus lunatus* 30.3 and *P. vulgaris* 63 | [41] |
| Soybean protein | - | Sequential anion-exchange chromatography, UF, reversed-phase chromatography | Anticancer | Lunasin | -Yield: 442 mg kg$^{-1}$ defatted flour (99% purity) -Histone-binding assays: biological activity is similar to those of the synthetic drugs | [42] |
| Soy protein | Neutrase | Sequential UF by spiral-wound (30, 10, 1 kDa) | Antiadipogenic | Peptides 1.3–2.2 kDa | Decrease of glycerol-3-phosphate dehydrogenase (GPDH) activity decreased (280 to 100 U mg$^{-1}$) and relative lipid accumulation (RLA) in 3T3-L1 cells | [43] |
| Corn protein | Alcalase | Sequential UF by Prep/Scale Tangential Flow Filtratuin (TFF) system 230V (Millipore) (RC 5, 3, 1 kDa) | Alcohol metabolism stimulator | Peptides < 1 kDa | -Activating hepatic alcohol dehydrogenase (ADH) -Peptides < 5 kDa triggered maximum ability to decrease blood alcohol concentration (BAC) | [44] |
| Corn protein | Validase fungal protease (FP), alkaline protease (AP), and/or neutral protease (NP) | Sequential UF by Millipore 8050 ultra-filtration (RC 10, 3, 1 kDa) | Antioxidants | F1: >10 kDa F2: 3–10 kDa F3: 1–3 kDa F4: <1 kDa | -NP-F3 had the highest activity (ORAC and Trolox equivalents, lipid oxidation inhibition) -AP-F2 and AP-F3 had highest activity (DPPH) | [45] |
| Broken rice protein | Alkaline protease | UF-coupled hydrolysis with hollow fiber module (Polyvinylidene fluoride (PVDF) 6 kDa) | Antioxidant | Oligopeptides (1.2–0.2 kDa) | -Oligopeptides content increased (60%) compared to batch hydrolysis (40%). 75% DPPH activity for 3-h operation | [46] |

**Table 2.** Significant UF-assisted separation process of protein hydrolysates with expressed bioactivities from marine sources.

| Source | Hydrolysis Enzyme(s) | Protocols | Bioactivity | Target Compound(s) | Remarks | Ref |
|---|---|---|---|---|---|---|
| Catfish muscle protein | Thermolysin | Sequential UF 10, 5 (Vivaflow 200), 3 kDa (Vivaflow 50) | ACE inhibitors | Peptides: GPPP and IEKPP | -GPPP (IC$_{50}$: 0.86 µm) -IEKPP (IC$_{50}$: 1.2 µm) | [47] |
| Yellow croaker | Neutral protease | Sequential UF 10, 5 (Vivaflow 200), 3 kDa (Vivaflow 50) | Antioxidant | Peptides < 3 kDa: SRCHV and PEHW | IC$_{50}$: 7.67 mg mL$^{-1}$ (O$_2^-$ scavenging) IC$_{50}$: 7.68 mg mL$^{-1}$ (DPPH) | [48] |
| Round cod muscle protein | Alcalase | Labscale TFF System, Millipore (10 and 5 kDa) | Antioxidant | Peptides < 5 kDa | Activity of sequence HDHPVC (0.7 kDa): IC$_{50}$ 0.38 mM (O$_2^-$ scavenging), 0.03 mM (DPPH) and HEKVC (0.6 kDa): IC$_{50}$ 0.37 mM (O$_2^-$ scavenging), 0.07 mM (DPPH) | [49] |
| Cuttlefish wastewater | Alcalase | Cascaded UF-DF Prep/Scale-TFF cartridges, Millipore (PES 100, 30, 10 kDa, RC 1 kDa) | -Antihypertensive -Antioxidant | Peptides < 10 kDa | Highest antihypertensive fractions from boiling water effluents by UF configurations: 100-30-10kDa (IC$_{50}$: 100 µg mL$^{-1}$). Highest antioxidant activities from (1) softening effluents by one UF filtration of 10 kDa PES (β-carotene assay: 9 µm BHT mg$^{-1}$, DPPH assay: 167 µm BHT mg$^{-1}$) and (2) from boiling water effluents by configuration 100-10 kDa (β-carotene assay: 18 µm BHT mg$^{-1}$, DPPH assay: 89 µm BHT mg$^{-1}$) | [50] |
| Yellowfin tuna's viscera | Protamex | Sequential UF by Amicon-stirred UF cell (RC 30, 10, and 3 kDa) | -Antibacterial -Antioxidant | Peptides < 3 kDa | -Minimum inhibition: 0.5 mg mL$^{-1}$, IC$_{50}$: 1.8 mg mL$^{-1}$ (DPPH); 1.4 mg mL$^{-1}$ (ABTS). Toward Gram—(*Escherichia coli*, 99.7% inhibition; *Pseudomonas aeruginosa*, 99.76% inhibition). Toward Gram + (*Listeria monocytogenes*, 99.88% inhibition; *Staphylococcus aureus*, 99.76% inhibition) | [51] |
| Hoki frame | Pepsin | Sequential UF (5, 3, 1 kDa) | Antioxidant | Peptides 1–3 kDa | 85% DPPH, hydroxyl radical, and alkyl radical scavenging activity in linoleic emulsion | [52] |

**Table 2.** *Cont.*

| Source | Hydrolysis Enzyme(s) | Protocols | Bioactivity | Target Compound(s) | Remarks | Ref |
|---|---|---|---|---|---|---|
| Argentine croaker | Flavourzyme/ α-chymotrypsin | UF cell (Advantec, UHP-76) (RC 0.5 and 1.0 kDa) | Antioxidant | Peptides higher than 1 kDa | Highest activity of lipid oxidation inhibition generated from Argentine croaker by Flavourzyme owing to greater proportion of hydrophobic sulfuric amino acids | [53] |
| Tuna dark muscle protein | Alcalase | Tangential UF-NF UF: tubular ceramic membrane 8 kDa NF: flat PES 1 kDa | Antioxidant | Peptides < 1 kDa Peptides 1–4 kDa | Peptides < 1 kDa: highest scavenging activity of free radicals (75%) and hydroxyl radicals (65%). Peptides 1–4 kDa: stronger iron chelating ability (40%) and highest superoxide radical (65%) | [54] |

**Table 3.** Significant UF-assisted separation process of protein hydrolysates with expressed bioactivities from dairy sources.

| Source | Hydrolysis Enzyme(s) | Protocols | Bioactivity | Target Compound(s) | Remarks | Ref |
|---|---|---|---|---|---|---|
| Egg white protein (US-pretreated at 40 kHz, 15 min) | Alcalase | Sequential UF by Millipore UF stirred cell unit 8050 (RC 30, 10, and 1 kDa) | Antioxidant | Peptides 1–10 kDa | Activity determined by DPPH: 28.10% ± 1.38% and ABTS: 79.44% ± 2.31% | [55] |
| Dephosphorylated egg-yolk protein | Alcalase, protease N, trypsin VI | Sequential UF (RC 5 and 1 kDa) | Antioxidant | Peptides < 1 kDa (amino acids H, M, L, F) | A 3-fold activity increase from nonhydrolysed dephosphorylated egg-yolk protein, confirmed by ORAC | [56] |
| Dilapidated egg yolk protein concentrates | - | UF-DF: (Spiral-wound PES 10 and 30 kDa) | Antioxidant | Phosvitin | Protein increased 6.25-fold by 10 kDa membrane (5.92-fold by 30 kDa). 10 and 30 kDaMWCO yields were similar (84%) | [57] |
| Goat milk protein hydrolysates | Subtilisin and trypsin | (1) casein and whey separated using 0.14 μm ceramic membrane; (2) retentate containing casein is hydrolysed; (3) hydrolysates then filtered by 50 kDa ceramic membrane | ACE inhibitors | Peptides < 50 kDa | Permeate of 50 kDa UF showed 30% enhancement of ACE-inhibitory activities (IC$_{50}$: 218.50 μg mL$^{-1}$) | [58] |

## 2. Advancements in Ultrafiltration for Protein and Peptide Separation

Fouling and concentration polarization are two major challenges in membrane filtration process which in turn affect permeate flux (productivity) and selectivity (purity) [59,60]. A comprehensive overview for preventing and mitigating these problems in wide range of pressure-driven membrane filtration has been reported [61]. In this paper, we focus on novel technologies of UF, with special attention in protein fractionation or isolation. The glimpse of optimizing operating conditions is delivered in Table 4. Meanwhile, pretreatment of feed and cleaning procedures are not the scope of this work.

### 2.1. Charged-Membrane

UF is a pressure-driven membrane process that uses a pressure difference to drive mass transfer through a porous membrane for separating solutes and solvents in the feed solution. The rate of solvent permeation trough the semipermeable membrane, or permeate flux (L m$^{-2}$ h$^{-1}$), is generally defined by

$$J_v = L_p \, dP/dx \tag{1}$$

where $L_p$ solvent permeability (L m m$^{-2}$ h$^{-1}$ bar$^{-1}$), $dP$ is TMP (bar), and $dx$ is the membrane thickness (m). The ability of the membrane to separate the solutes, or sieving coefficient $S_o$, is expressed as [62]

$$S_o = c_p/c_r \tag{2}$$

Here, $c_p$ and $c_r$ are concentrations of solute in permeate and retentate, respectively.

Solutes that are retained by the membrane build up on the membrane surface and create a new layer. The presence of a new layer leads to increased mass transfer resistance. Sometimes, this layer becomes a new filter that has a smaller aperture than the membrane itself. The concentration of solute in the membrane wall, $c_w$ can be estimated by [62]

$$c_w = c_b \left\{ S_o + (1 - S_o) \exp \left( J_v/k \right) \right\} \tag{3}$$

Equation (3) shows that $c_w$ depends on concentration of solute in the bulk solution ($c_b$), sieving coefficient ($S_o$), permeate flux ($J_v$), and mass transfer coefficient at the boundary layer ($k$).

Charged-membrane UF is used to address the drawbacks of size-based sieving owing to the electrostatic repulsion or rule-of-thumb of at least a factor of 10 in molecular mass difference for complete separation [63]. It can be seen in the exemplified performances of the charged UF membrane in protein separations, tabulated in Table 4, that performance of the charged UF membrane improved by more than 50% of that of pristine-membrane UF. The advantage of using a charged membrane are implied in the flexible design of membrane and exemplified in the elimination of nearly complete protein in permeate to produce whey protein concentrate, employing the polarization index ($\beta$) constant, which is defined by [62]:

$$\beta = \frac{J_v}{k} = \frac{J_v}{0.816 \left( \frac{6 \, Q_R}{h \, V_{HR}} D^2 \right)^{1/3}} \tag{4}$$

where $Q_R$ = retention flow rate or average recirculation rate, $h$ = spacer height; $V_{HR}$ = retentate hold-up volume, and $D$ = protein diffusion coefficient $\approx 2 \times 10^{-7}$ cm$^2$ s$^{-1}$.

Since parameter $\beta$ represents the membrane system design, scale-up with the same performance can be achieved with different materials, module geometry, and membrane area, provided that $\beta$ is set constant (in order to maintain stable thickness of deposit layer). At the same time, controlling $\beta$ will stabilize the flux, and hence increase membrane lifetime due to minimization of cleaning-in-place [64]. This stable and predictable target would be rather unlikely to achieve if TMP is held as reference as in normal practice in dairy industries [65,66].

Mehta and Zydney [67] reported a study of the effect of membrane charge on permeate flux and protein transport through the UF membrane. The actual sieving coefficient, $S_a$, for a charged UF membrane is estimated by the following equation

$$S_a = (1 - r_s/r_p)^2 \, K_c \exp E_s \tag{5}$$

where $r_s$ and $r_p$ are radii for solute and membrane pores, $Kc$ is the convection hindrance factor, and $Es$ is the dimensionless interaction electrostatic energy. $E_s$ depends on ionic strength of the feed solution, membrane pore radius, and the particle size or radius of solutes. Equation (5) shows that the transmission of protein decreases with the increasing membrane potential, which is in agreement with the experimental results. Charge on the membrane surface creates electrostatic interaction with protein improving the sieving coefficient.

Membrane UF can be charged with positive or negative moieties, and the details of preparation were published elsewhere [68]. However, mechanistic explanation of protein transmission is limited, excluding a study to correlate stagnant film model in charged UF membrane for multicomponent mixtures [69]. Therefore, a study with mechanistic approach to optimize the charged-membrane UF would be interesting to expand in the future.

**Table 4.** Representative works of protein ultrafiltration advancements.

| Process | Feeds | Membrane and/or Protocols | Remarks | Ref |
|---|---|---|---|---|
| UF with charged membrane | Whey protein | Regenerated cellulose (RC), negatively charged 100 kDa (flat sheet/spiral-wound) | Both modules retain 98% of total whey proteins 85% higher flux than 10 kDa pristine membrane | [62] |
| | $\alpha$-lactalbumin and $\beta$-lactoglobulin | RC, positively charged 300 kDa (two-staged UF) | 490% improvement of selectivity to retain $\beta$-lactoglobulin. Purity of $\alpha$-lactalbumin: 87% | [70] |
| | Milk serum permeate | RC, positively charged 300 kDa (three-staged UF) | 180% improvement of selectivity to retain $\beta$-lactoglobulin. Purity of $\alpha$-lactalbumin: 87% | [71] |
| | Lysozyme and lactoferrin | Zirconia, positively grafted 300 kDa by ethylenediamine | Lysozyme transmission selectivity up to 165 100% purity in permeate | [72] |
| EUF | BSA | Continuous current, PS (polysulfone) 100 kDa, effective area 45 cm$^2$, 3000 V m$^{-1}$ | Reduction of 80% in BSA concentration time (from 0.5–1.0 g L$^{-1}$) | [73] |
| | BSA and lysozyme | External DC current, PES (polyethersulfone) 30 kDa, effective filtration area: 132 cm$^2$, 1000 V m$^{-1}$ | Decline of lysozyme retention by 53% Permeate flux increased around 23.4–36.7 L m$^{-2}$ h$^{-1}$ | [74] |
| | | Pulsed electric fields, fluoride polyvinylden flat sheet 25 kDa, effective filtration area: 51 cm$^2$, 700 V m$^{-1}$ | A 300% increase in permeate flux Reduction of membrane abrasion | [75] |
| | BSA and lysozyme | Electrodialysis integration, UF membrane: PES 30 kDa, effective filtration area: 35 cm$^2$, 150 mA | Permeate flux increase by 20% while operated in high conductivity feed | [76] |
| EDUF | $\alpha$-lactalbumin and $\beta$-lactoglobulin | CA, 100 kDa (elution mode), membrane active area: 35 cm$^2$, 3–22 A m$^{-2}$, 100–800 V m$^{-1}$ | Separation factor of 1.2 reached at pH 4.8 (nearby IEP of $\alpha$-lactalbumin) (optimum) | [77] |

**Table 4.** *Cont.*

| Process | Feeds | Membrane and/or Protocols | Remarks | Ref |
|---|---|---|---|---|
| | $\alpha$-lactalbumin and bovine hemoglobin | CA (100 kDa), membrane active area: 32 cm$^2$, 3–20 A m$^{-2}$, 170–800 V m$^{-1}$ | Elution mode: 99% purity of $\alpha$-lactalbumin Separation mode: 20% increase in yield | [78] |
| | Herring milt hydrolysate | Electrocell unit (membrane active area: 10 cm$^2$), UF membrane 20 kDa, 11 Vcm$^{-1}$ | 214% increase in antioxidant activity | [79] |
| EDBM | Soy protein hydrolysate | Electrocell AB unit (eff electrode area: 100 cm$^2$) UF integration (hydrophilic RC 100 kDa, 13.4 cm$^2$) | Soy protein concentrate 70% with phytate below 1% Eliminate clogging, but not fouling | [80] |
| | Lactose-enriched hydrolysate | New 6-compartment EDBM, Electrocell AB unit (eff electrode area: 100 cm$^2$), UF membrane: spiral-wound 10 kDa (area 2.13 m$^2$) | Complete reduction of scaling | [81] |
| Ultrasonic-assisted UF | Whey protein concentrate | PES 30 kDa and permanently hydrophilic PES 5 kDa (effective area 100 cm$^2$), 20 kHz, 300 W (low frequency) | Reduce irreversible fouling and increase cleaning efficiency up to 17.23% (5 kDa) and 5.47% (30 kDa) | [82] |
| Dynamic UF | Mungbean wastewater | RDM: ceramic (MgAl$_2$O$_4$) 15 kDa (eff membrane area: 360 cm$^2$), TMP 1.2 bar, 1403 rpm, and VRF 5.0 (optimum) | Protein recovery by tenfold higher flux at 42 L m$^{-2}$ h$^{-1}$ than conventional cross-flow UF. Protein retention: 96% | [83] |
| | UHT milk | VSEP: PES 10 kDa, effective area: 503 cm$^2$, 60.75 Hz, 45 °C | Operation at higher VRF (9.0) than cross-flow UF with nominal commercial flux (20 L m$^{-2}$ h$^{-1}$) | [84] |
| HPTFF | Chicken egg white | 30 kDa PES, 2100 rpm, pH 10–11, and NaCl 100 mM | Lysozyme–ovalbumin (different in size and charge) attained high resolution (transmission: 99%, selectivity: 2400; flux of 25.6 L m$^{-2}$ h$^{-1}$) | [85] |
| | Monoclonal antibody Alemtuzumab | PES 300 and 100 kDa, PVDF 100 kDa constant flux: 7.368 × 10$^{-6}$ m s$^{-1}$ | Only PVDF 100 kDa gave 93.3% purity for monomer (155 kDa) after one-staged diafiltration | [86] |

VRF, volume reduction factor $= \frac{V_0}{V_0 - V_a - V_p}$, where $V_0$ is initial feed volume, $V_a$ is average permeate over filtration, and $V_p$ is final permeate volume; EUF = electroultrafiltration; EDUF = electrodialysis with ultrafiltration; EDBM = electrodialysis with bipolar membrane; HPTFF = high-performance tangential flow filtration.

## 2.2. Electroultrafiltration (EUF) and Electrodialysis with UF Membrane (EDUF)

While separating specific protein fractions, concentration polarization may hinder the full potential of an operation unit. To tackle this issue, mechanical interventions require complex control system, but are harder to scale up. Another approach exists which is to elaborate the electrokinetic phenomena comprising of electrophoresis, electroosmosis, and electrolysis into ultrafiltration, namely electroultrafiltration (EUF). Overall, EUF could provide an intriguing basis to ameliorate concentration polarization and foulant deposition without applying shear rate in UF owing to detachment of charged molecules in the form of a gel layer from the membrane surface [76].

In EUF, the maximum component flux at given TMP, $J_{max}$, can be estimated by [87]:

$$J_{max} = \mu e \, E_c \tag{6}$$

Here, the maximum flux depends on critical electrical potential ($E_c$) and the electrophoretic mobility of the component ($\mu e$). The effect of electrical field strength is then introduced to the gel layer model, leading to a new equation for estimating the flux of molecule toward the UF membrane [87]:

$$J = k\{(c_w - c_p)/(c_b - c_p)\} + \mu e\,E \tag{7}$$

Electrophoretic effect mainly governs EUF along with ion association, ion adsorption, or ion dissolution and serves the gradient of electrical field to induce protein mobility. Meanwhile, the movement of solution through porous material under this driving force is called electroosmosis. Although electroosmosis increases the permeate flux, this does not chiefly regulate the process [88,89]. In contrast, electrolysis is a breakdown phenomenon of organics under electrically driven chemical reaction on the electrodes.

Gradiflow Technology arranges the EUF process with parallel electrodes (cathode and anode), which are overflowed by continuous buffer solution and electrified. In the center of the apparatus, the UF membrane is attached between two restriction membranes to start the electrical gradient in the same direction of voltage going in a perpendicular vector with feed flow. Thus, protein mobility will shift between opposite electrodes. In order to obviate the probability of proteins being denatured by electrolysis and protein deposition on the electrodes, electrodes must not be placed inside the suspension and its permeate entry or exit [90].

Some selected works of protein separation by EUF are summarized in Table 4. EUF accelerated bovine serum albumin (BSA) concentration by 80% with bigger molecular weight cutoff (MWCO) of a polysulfone membrane to concentrate 0.5 g L$^{-1}$ BSA solution to 1.0 g L$^{-1}$ [73]. Again, with two-sided EUF, not only can high purity and short separation be achieved, but also selectivity of fractionation of binary proteins, i.e., BSA and lysozyme. It was said that the selectivity was enhanced to be above 800, owing to the electrophoretic effects that allowed the filtration velocity to be kept high for a prolonged amount of time [91]. It is interesting to note that the environmental conditions of the solution and the membrane configurations can influence the performance in an agonistic or antagonistic way during EUF. For instance, during the separation of BSA and lysozyme at constant external direct current (DC) electrical field of 1000 V m$^{-1}$ and pH 7.4 (between the IEP (isoelectric point) of BSA of 4.7 and lysozyme of 11.0), permeate flux increase between 23.4 and 36.7 L m$^{-2}$ h$^{-1}$ and retention of lysozyme declined by 53% from 73% [74].

Despite all of those advantages, only proteins that have wide difference of IEP can be separated successfully. However, EUF suffers from productivity limitation only in the level of milligrams per hour. Furthermore, EUF can impose conflicting consequences on performance if operated under high conductivity because of the divided electric field between protein and electrolyte mobility [19]. Therefore, in conventional EUF, continuous current is applied; that consumes high energy requirement due to the heat production and alteration of protein on account of electrolysis and leads to other problems. In order to tackle these challenges, the latest development tailors EUF with intermittent or pulsed electric-field (PEF) UF, where the restoration of permeate flux was found to be better. One study showed that turbulence initiated by the static metal sheet, formation of oxygen bubbles around the membrane skin, and the electrophoretic effect by the pulsed electric field at electric field intensity (E) of 700 V m$^{-1}$ provided a 300% increase in permeate flux, parallel to the reduction of membrane abrasion [75]. Higher voltage (until it reaches the critical voltage) and shorter voltage of both pulse interval and pulse duration delivered the highest permeate flux and fully mitigated the fouling, which were useful in membrane cleaning [92]. Here, about 25–40% decrease in resistance can be attained by PEF as opposed to the conventional cross-flow UF with zero electric field [93]. Furthermore, PEF treatment has been investigated to improve antioxidant properties of protein hydrolysates of low molecular weight (10–30 kDa) peptides from egg white antioxidant activity, which increased by 44.23% at 10 kV cm$^{-1}$ with a pulse frequency of 2000 Hz [94].

Electrodialysis with UF membrane, abbreviated as EDUF, is another strategy to address these deficiencies by uniting the ion-exchange membrane, UF membrane, and electric field between the cathode and anode. Electrodialysis (ED) utilizes electric potential difference and ion-selective membranes (cation-exchange membrane, CEM and anion-exchange membrane, AEM) for ionic separation [95–97]. Removal of ions is governed by

$$c_f - c_d = I/(z\, F\, Q) \tag{8}$$

The removal rate or concentration difference between diluate or product ($c_d$) and feed ($c_f$) is determined by applied current, feed flowrate ($Q$), ionic valence $z$, and Faraday constant $F$. Only ionic substances are transported in through ion-exchange membranes in ED, while solvent is almost unaffected by the electrical field. CEM is permeable for cations while excluding anions. In contrast, AEM allows the permeation of anions but rejects cations. The introduction of UF into the ED module allows one to separate ionic components that have different molecular weights. EDUF can separate salt and protein into different compartments [76], and may be employed within either separation or elution modes as depicted in Figure 2.

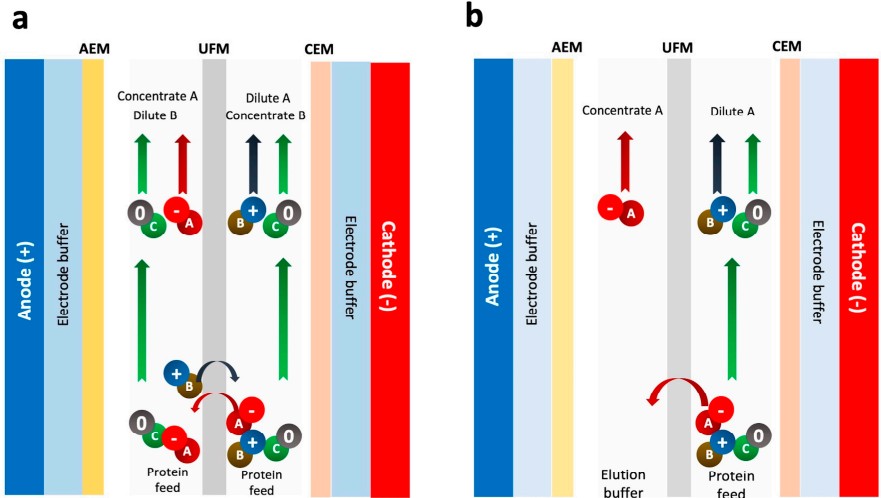

**Figure 2.** Electrodialysis with UF membrane (EDUF) in (**a**) separation and (**b**) elution mode. CEM: cation-exchange membrane; AEM: anion-exchange membrane; UFM: UF membrane.

Separation mode can be used to attain high productivity, although with the cost of not-too-selective isolation. Since multiple feeds are flowed through, this will divide the protein based on charge, each on concentrated and diluted side. In Figure 2a, negatively charged A is the primary target and positively charged B is the secondary target, making CEM embrace the concentrated (A-rich) stream and AEM embrace the diluted (B-rich) stream. The elution mode aims to achieve higher purification, but that requires the sacrifice of its productivity. So, as only one feed is passed through, the majority of targeted compounds would be collected in either the diluted or concentrated stream. In Figure 2b, owing to the targeted and negatively charged A, CEM interface will directly be accessible toward the concentrated (rich A) stream while AEM faces lower concentration of both A and B (compared to separation mode). In an experiment to isolate α-lactalbumin and bovine hemoglobin, at initial concentration of 0.1 g L$^{-1}$, 99% of α-lactalbumin purity was obtained at concentrate side by elution mode. Meanwhile, superior yield by separation mode exceeds the elution mode by 20% [78]. Promisingly, EDUF can enhance the bioactivity of isolate like portrayed in fractionating herring milt hydrolysate that attained 214%-increase in antioxidant activity [98], and 125 μg mL$^{-1}$ of 0.3–0.5 kDa cationic peptides exhibit in vitro noncompetitive inhibition toward ACE and dipeptidyl peptidase IV (DPP-IV) after single-step 4-h treatment [79].

In order to increase the purity and selectivity, in particular, permeate flux to compete that of UF with the same membrane and length of the process, configuration of EDUF can be arranged as disclosed in another review [32]. Hitherto, there are six stand-out configurations relevant in EDUF practices, which are described in Figure 3a–f, of which configuration (a) is the most popular. Furthermore, integration of EDUF with other UF process and physical pretreatment of feeds are sought to grow the acceptance of EDUF in industries. First, when treating soy protein isolate, operation of EDUF at pH 3.0 and 6.0 amplified the antioxidant capacity owing to higher concentration polar peptides with lysine moieties (0.4–0.5 kDa) from anionic compartment compared to the NF when tested in vitro against reactive oxygen species by human neuroblastoma cell line. Of isolates generated from NF and EDUF at pH 9.0, only the capability of degrading $H_2O_2$ was observed. Thus, sequence integration of EDUF-NF would be advantageous to refine the UF permeate [99]. Second, pretreatment of defatted flaxseed protein isolate with high hydrostatic pressure (HHP) at 400 MPa HHP and 21 °C for 20 min, and further processing with EDUF resulted in isolate (recovered in KCl) rich in arginine, which lowers systolic blood pressure [100]. Arginine itself cannot be separated using conventional electrodialysis since it adulterated the CEM in basic EDUF [101].

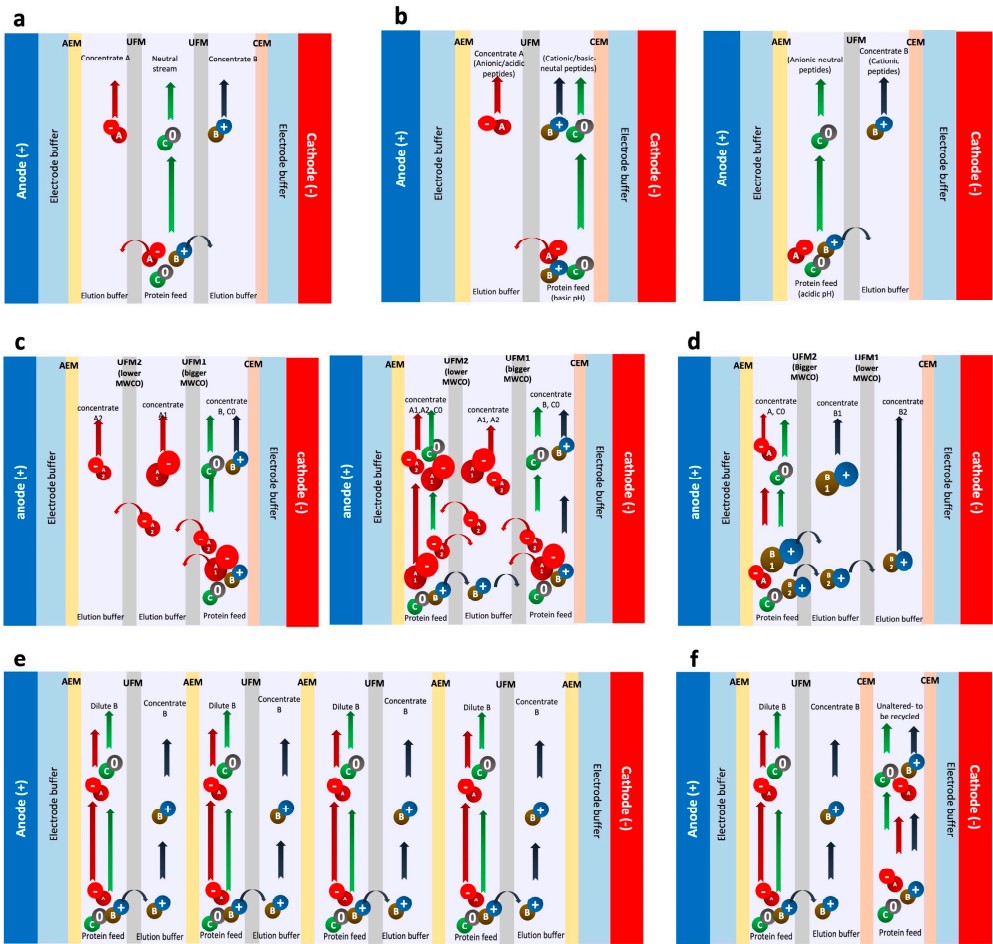

**Figure 3.** Electrodialysis with UF membrane (EDUF) with six different configurations for (**a**) parallel fractionation of cationic and anionic peptides (single feed); (**b**) cationic and anionic peptides fractionation at basic (left) and acidic pH of feeds (right), respectively (single feed); (**c**) series fractionation by double UFM for anionic peptides in different compartment with single feed (left) or for anionic peptides in single compartment with dual feeds (right); (**d**) series fractionation by double UFM for cationic peptides in different compartment (single feed); (**e**) alternating anion-exchange membrane and UFM for acidic peptides fractionation (multiple feeds); (**f**) double CEM for cationic peptides fractionation (double feeds). CEM: cation-exchange membrane; A or AEM: anion-exchange membrane; UFM: UF membrane.

Amidst all of the advantages in operating EDUF, water splitting, fouling, and vulnerable membrane integrity still obstruct the scale-up strategies. Despite having minor degradation in UFM, the protein fouling on the ion-exchange membrane cannot be ignored because the ion-exchange membrane is a nanoporous membrane [102]. In the same work, after six consecutive fractionations, a more debilitating AEM as resulted by electrochemical degradation (showed by increase in roughness) was detected since its diluted side generated a great amount of OH$^-$ ions. This phenomenon also turned up while EDUF operated at high current densities, provoking extreme basic solution in favor of alkaline hydrolysis, thermal degradation, and electrochemical deterioration [103]. Although generally H$^+$ and OH$^-$ ions are developed by water splitting, the emergence of concentration polarization would expedite the process of coming across "limiting current density" on diluted side of ion-exchange membrane [104]. In this context, ion concentration is close to zero, so both H$^+$ and OH$^-$ ions' mobility may shift away from the diluted part (electroconvection), which will reduce concentration polarization in one side, but will contrarily digress pH in all compartments and curtail membrane selectivity [32,105].

### 2.3. Combining Electrodialysis with Bipolar Membrane (EDBM) and UF

Electrodialysis with bipolar membrane (EDBM) is conducted in a similar concept with EDUF, but with the objective of electroacidification by adjusting the H$^+$ and OH$^-$ ions, generated from water splitting in anionic and cationic compartments using controlled current. In BM, AEM and CEM are laminated into a single membrane. The bipolar interface created by AEM and CEM forms a catalytic layer that induces an extensive water dissociation reaction when an electrical potential is applied to the electrode of EDBM. The free energy required, $\Delta G$, for the water dissociation reaction is expressed as [106]:

$$\Delta G = F\, \Delta\varphi = 2.3\, R\, T\, \Delta pH \tag{9}$$

where $F$ is the Faraday constant, $\Delta\varphi$ is the electrical potential difference between two solutions separated by the BM, $R$ is the gas constant, $T$ is the absolute temperature, and $\Delta pH$ is the pH value difference between two solutions separated by BM.

To illustrate, during dairy electroacidification, NaOH is liberated from the compartments of EDBM cells and can later be recycled for resolubilization. So, the minimum chemical is used compared to conventional acid precipitation that requires NaOH addition. Moreover, most of the coproduct from the base stream can be reutilized for caseinates [107]. With respect to other positive features such as increased protein content and better removal of antinutrients and minerals (calcium and magnesium), the performance of EDBM in dairy usage is commonly hampered by: (1) caseins clogging on spacers inside the EDBM stack and protein fouling on the bipolar membrane (BM) and cation-exchange membrane (CEM) and (2) scaling (mineral fouling) on the interface of the CEM [108,109], which lead to low permeate flux and longer filtration time [110].

The clogging issue was successfully been addressed by coupling EDBM with UF. UF was used for treating input of hydrolysates or protein solution in a way that allows protein retention before electroacidification in EDBM stack. This method was capable of achieving thorough elimination of protein precipitation in dairy processing [111]. In soy protein concentrates consisting of 70% minimum protein content, despite encountering fouling, EDBM-UF produced low phytate content (lower than 1% $w/v$) and enhanced productivity and purity on account of demineralization [80]. In order to tackle persistent fouling problem, one study tried to incorporate discontinuous diafiltration (DDF), which was able to isolate higher protein concentration by minimizing carbohydrates and minerals. However, this failed to repair the problems in permeate flux of electroacidified feed [80].

Recently, treatment of soy protein extract at pH 6.0 with EDBM-coupled UF/DF by 100 kDa membrane was furthermore found to be best at producing isolate with the lowest phytic acid and increased solubility in a pH range of 2.0–4.0, compared to isoelectric precipitation at pH 4.5 and integrated UF/DF at pH 9.0 [112]. Researchers extended the exploration of how the sequence of embedded UF/DF process can be essential to the

overall performance of integrated EDBM-UF/DF [113]. From all variation to attain final permeate volume 1.5–1.6 of feed, UF with VRF5 and continuous diafiltration (CDF) with VD 4 (volume diluted) suggested not only the most severe fouling and lowest permeate flux, but also the highest removal performances of phytic acid and carbohydrates, which were more than (1) UF-discontinuous diafiltration (DDF) with VRF 5 and re-VRF 5, (2) UF-CDF with VRF 2 and VD 2, and (3) UF-CDF with VRF 2 and double VRF 2 [113].

Various research studies explored the solution of another EDBM challenge, which is scaling on the CEM interface with its alkaline concentrate side owing to the facilitated mineral precipitation (see Figure 4) that set up the decrease of membrane permselectivity and increased global resistance [114]. First, they performed the electroacidification of milk until reaching pH 5.0, not at 4.6 on the ground of lower migration of free $Ca^{2+}$ and $Mg^{2+}$ since the remains of these ions are associated with casein micellar and higher $K^+$ transfer [115]. This protocol resulted in 27%-decrease in CEM scaling [111]. Second, the addition of pulse electric field (PEF) deprived the CEM scaling by perturbing the growth of $Ca^{2+}$ and $Mg^{2+}$ deposition [116], and ultimately decreased the scaling to the point of 40% [111]. Third, hydrodynamics of EDBM compartment was modified by increasing the input flow rate. Furthermore, additional $K^+$ was inserted into the compartment by additional KCl with aim to inhibit the $Ca^{2+}$ and $Mg^{2+}$ mobility to basic compartment, and thereafter scaling on CEM surface (Figure 4b). Each alternative reflected 30% and 38% reduction of scaling, respectively [117]. Lastly, configuration of EDBM cell was found to determine the overall performance. Adding CEM into conventional five compartments EDBM (Figure 4c) would inhibit the passage of $OH^-$ from BM into the concentrate stream ($Ca^{2+}$ and $Mg^{2+}$ enriched solution) [81]. In contrast, $H^+$ can strongly follow the new CEM and sustain acidic pH (4.0) for the first 20 min of operation. The introduction of a new compartment successfully avoided scaling formation by preventing $OH^-$ migration to the CEM (Figure 4d). Generally speaking, the combination of EDBM-UF integration and complementary stacking of double CEMs side-by-side with the BM unraveled the procedure to create complete scaling and clogging prevention in EDBM.

## *2.4. Ultrasonic-Assisted UF*

Ultrasound (US) has been used to aid broad spectra of separation process like feed pretreatment [118–121], enzymatic hydrolysis [55], extraction, and centrifugation [122,123], including membrane UF [82,124–126], with special attention toward reduction of fouling and concentration polarization. By transferring considerable mechanical power through small mechanical movements from liquid (high intensity) into a gaseous medium, US can work to disrupt the structure of the cake layer and concentration polarization covering the surface of membrane. US transition and propagation are comprised of two foremost phenomena categories, which are (1) cavitation (liquid jets), radiation pressure, and acoustic streaming and (2) physicochemical transformation (dispersion, coagulation, and liquid property changes) [127].

To illustrate the performance of US, pretreatment of egg white protein (EWP) by US and subsequent UF was conveyed to endow more antioxidant features (measured by DPPH and ABTS methods) of low molecular weight peptides (<1 kDa as much as 11.19 ± 0.53%, and 1–10 kDa for 28.80 ± 0.07%) [55]. Likewise, whey protein concentrate pretreated shortly (below 5 min at 20 kHz) by US showed reduction in viscosity owing to smaller aggregates [128], and advanced heat stability [129]. In treating cheese whey wastewater with UF, ultrasonication was used as a method of reducing chemicals for membrane cleaning [82]. It is also shown in another work that the treatment with ultrasonication (20 kHz, power 300 W) was more effective for irreversible fouling, which gave a 17.23% increase in cleaning efficiency of permanently hydrophilic PES as opposed to 5.47% in PES (30 kDa). In the end, profitability analysis provided the data that only small proportion of additional cost was needed for US in comparison with NaOH treatment [82].

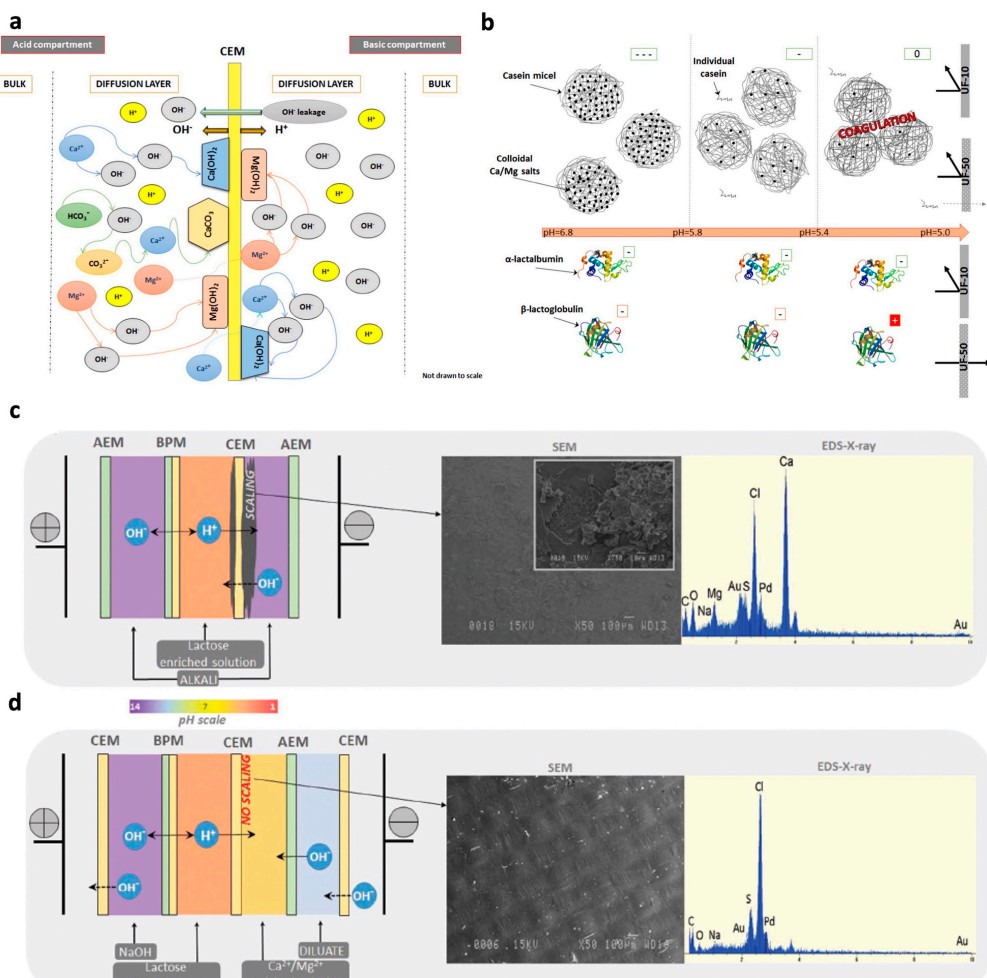

**Figure 4.** Scaling and fouling in EDUF during protein separation. (**a**) Mechanism of induction of major scaling agents: Ca(OH)$_2$, CaCO$_3$, and Mg(OH)$_2$. (**b**) Plausible transport of casein, α-lactalbumin, and β-lactoglobulin in pH 6.8–5.0. Membrane with MWCO larger than 10 kDa is not recommended for this procedure since whey proteins and individual caseins may penetrate through the CEM and cause organic fouling. (**c**) Conventional five-compartments with represented scaling on CEM. (**d**) New 6-compartments without observed scaling on CEM. Panels (**a**) and (**b**) are reproduced from [117] with permission, © 2021 Elsevier Ltd. Panels (**c**) and (**d**) are reprinted from [81] with permission, © 2021 Royal Society of Chemistry).

Some research studies have shown that better cleaning performances can be obtained from lower frequencies of US and higher power intensity since high intensity implies in the detriment of membrane material [130] and protein denaturation [131]. However, the collapse of larger bubbles generated from acoustic cavitation at low US frequencies resulted in microjetting and turbulence localization, insomuch as the less intense of cavitation collapse from smaller bubbles at higher frequencies [132]. Indeed, low US frequency has better cleaning efficiency and flux performance, but a slight protein denaturation can still inflict in the form of self-aggregation with free casein micelles in the first 30 min of ultrasonication, even at low frequency (20 kHz) with power 20 and 41 W. After all, the viscosity of milk and casein micelles were not transformed by acoustic cavitation, and the size of fat globules decreased with ultrasonication at lower power [133].

Submerged module in US bath also had higher efficiency than placing US in the cleaning solution as a novel way of US membrane cleaning if protein is attached cohesively with membrane surface by cationic bonding due to calcium–protein interaction. Otherwise, it would not expose a different inclination in cleaning efficiency [125]. The cleaning efficiency is more related with the conditions of fouling solution. For instance, despite

substantial flux enhancement was reached at 20 kHz and 2 W cm$^{-2}$ power intensity, increasing protein concentration from 27 to 216 g L$^{-1}$ led to the evolution of rheological regimes from Newtonian to shear-thinning and further solid fouling deposits [134].

Evidently, membrane materials determined the performance of this process as suggested by superior performance in ceramic membrane than in polymeric membrane, and with added detergent (P3-Ultrasil 67 and 69) compared to alkali solutions (NaOH) for 30 min at 35 kHz [135]. A comprehensive review in explicating the effect of US on membrane material, however, showed that the membrane materials tested, polyvinylidene fluoride (PVDF) is in some aspects statistically claimed to be the most resistant toward ultrasonication [31]. In this case, the PVDF membrane was prepared by ultrasound-assisted phase inversion, and their increases in porosity and elongation were acquired by low US frequency and high US power (45 kHz, 300 W, 1 min) [136].

*2.5. Dynamic UF*

Dynamic UF aims to impede the development of fouling and concentration polarization by leveling up the shear rate [137], which thereafter stimulates turbulences from induced rotational speeds (centrifugal and radial drag forces), notably in mass-transfer-controlled regime [138,139]. This can be generated by two ways: (1) arrangement of relative motion by assembling the membrane with a rotating disk, impeller, or vibration system as a moving part and (2) developing self-rotating membrane UF. In one review, rotating cylindrical membranes, a rotating disk system, and a vibrating system were extensively discussed with regards to the design, mechanism, and application for general membrane filtration in MF, NF, UF, and reverse osmosis (RO) [33]. Conversely toward conventional filtration, in dynamic UF, the energy requirements and TMP reduction can be minimized since participation of tangential velocity that generates large axial pressure gradients is unnecessary [140]. The major advantages of dynamic filtration are more profound in low MWCO membrane as they will enable operation at high TMP and rise flux until 4000 kPa owing to the extended pressure-controlled regime and lower concentration polarization. Both promising concentration factor (CF) (defined as the ratio between concentration of the concentrate recovered after filtration to initial concentration of solution filtered) and membrane selectivity can be gained with highly viscous solution, and these are implicated in high water recovery [141,142]. Moreover, using the second generation of shear-enhancing modules like vibratory shear-enhanced processing (VSEP) or multishaft rotating membranes (RDM) and choosing ceramic (not polymeric) membranes, the performance of dynamic UF can be powerfully enhanced [33,143].

Applications of dynamic UF to reach high VRF are rapidly increasing; one of them is for protein recovery in mung bean wastewater [83]. At TMP 1.2 bar, rotating speed 1403 rpm, and VRF 5.0, they achieved tenfold higher flux at 42 L m$^{-2}$ h$^{-1}$ as opposed to the cross-flow system. Yet, stabilization (recycling back permeates to feed tank, and gradually increase TMP from 0.4 to 1.0 bar) was found not to positively contribute to fouling minimization and flux enhancement, but in fact, it prolonged the operation time. RDM application can be used to isolate therapeutic small peptides effectively near the IEP, and this has been developed for sesame protein hydrolysate, from which they comprehended that stirrer rotation was limited in use for generating suitable shear rate [144]. Numerous research studies applied rotating disk membrane to fractionate α-lactalbumin and β-lactoglobulin from whey solution. Since these proteins have similar molecular weight, standard dynamic UF can only see low yield and selectivity. For example, a work with α-lactalbumin transmission was stagnant between 0.2 and 0.13, the yields were from 28% to 34% [143].

Either rotating disk membranes [140,141,143,145–147] or vibrating membranes (VSEP) [84,148–151] have been intensively used for protein concentration or fractionation in several studies. On the opposite, self-rotating membrane, although it can supplement the complexity of conventional dynamic UF, to the best of our knowledge, is not well-known as an up-to-date application for UF. One experiment of Meyer et al. (2015) performed dy-

namic UF with self-rotating membrane for concentrating skim milk protein by a pilot-scale apparatus, and compared the result with spiral wound membrane (SWM) [152]. They described an intriguing phenomenon that there was a critical flux controlling permeate flux in dynamic UF by self-rotating membrane as medium rotational speed (1000 min$^{-1}$) gave the highest flux (35 L m$^{-2}$ h$^{-1}$) due to permeate backpressure or altercation of the smaller particle deposition beyond a critical speed. Flux was also found to increase linearly until TMP 100 kPa. Moreover, high VRF, even for more than 8.0, is not a primary factor that needs to be considered to deteriorate flux during dynamic filtration compared to SWM, which is only acceptable for VRF < 5–6.

### 2.6. High-Performance Tangential Flow Filtration (HPTFF)

Tangential velocity is used to prevent the foulant deposition on and in membrane pore. HPTFF basically belongs to the category of dynamic UF, except it also separates proteins based on net charge besides molecular size. Not only does it improve the performance, but HPTFF can also cater a new process design that is more economically practical yet discharges minor wastewater. Exploration of HPTFF has been used to simplify isolation methods by joining hydrophobic interaction chromatography and subsequent UF/DF in one unit. From this research, it was found that the performance of HPTFF was similar to conventional process, but the yield of Fab'2 from recombinant *E. coli* with 100 L m$^{-2}$ h$^{-1}$ was 12% higher than the UF/DF process, and there was no loss of selectivity after 10 times regeneration of novel positively charged 100-kDa RC, modified by bromo-propyl-trimethylammonium bromide [153].

In HPTFF, high selectivity can be achieved by (1) minimizing fouling and concentration polarization with operation in pressure-dependent regime just below the transition point, (2) increasing disparities of hydrodynamic volume or effective size (core dimension and its extended electric double layer) between products and byproducts by adjusting pH far from the IEP and optimum ionic strength (low value inhibits buffer ions generating the shielding of charges), (3) enhancing electrostatic repulsion of compounds with alike size, (4) signifying purification and yield by diafiltration, and (5) decreasing TMP with co-current filtrate flow configuration [154–157]. For molecules exhibiting radically different charge and/or size, HPTFF was proven as a promising advancement in UF for protein separation of lysozyme and ovalbumin from chicken egg white [158], BSA and hemoglobin [159], and lactose from pretreated casein whey [160].

For similar charges, HPTFF can hardly perform with notable yield and selectivity. For instance, size-based removal of peptide dimer or oligomer from monoclonal antibody alemtuzumab only showed monomer purification factor of four for one-staged diafiltration [86]. Recent strategy of improving purity and yield was by engineering molecular charge in order to arrange the electrostatic interaction and protein transport. One finding to obtain pure myoglobin demonstrated that retention of lysozyme was increased by the present of BSA (completely rejected by 30 kDa MWCO) as dual-facilitating agent since negatively charged lysozyme was attracted by positively charged BSA (IEP 4.9) at pH 9.0 and myoglobin migration to permeate will follow the Donnan Effect [161]. Small charge affinity ligand like Cibacron Blue has been used to manipulate BSA net charge [154,162,163]. In another research, small-stirred UF cell and negatively-charge regenerate cellulose (with covalent attachment of sulfonic acids) were employed to separate BSA and ovalbumin, and reached high resolution: 30-fold selectivity, 90-fold improvement of purification factor, and more than 90% yield of BSA [162].

By replicating the process for HPTFF (two stages diafiltration) with Pellicon XL tangential flow module which can be scaled-up linearly for membrane area above 80 m$^2$, the result attained more than 80% yield and 15-fold purification factor of BSA in retentate, whereas ovalbumin was of 95% yield with 50 L m$^{-2}$ h$^{-1}$ permeate. In this research, they suggested that DV number can be manipulated to elevate the yield and purification factor [154]. Despite those distinguished features, HPTFF is vulnerable to bigger MWCO membrane. This trend was confirmed by Zhang et al. that testing UF rotating disk membrane (UF-RDM)

for separation and concentration of leaf protein from alfalfa juice [23,145,146]. Higher MWCO and temperature was attributed to aggravating irreversible fouling and separation performance during VRF = 6 and VRF = 12, but also to leveraging better filtration behavior and productivity. However, HPTFF requires a high energy cost [71].

### 2.7. Integration of UF with Other Processes

In general, UF is easier to conduct with highly diluted hydrolysates (equal to or lower than 1% *w/v*) and low VRF. However, this will lead to an uneconomical process at the industrial scale due not only to productivity reduction, but also to a wider membrane area requirement. Several techniques of UF integration are also plausible to tackle this problem and explicated as follows:

A study showed the effect of two-step separations of fish protein hydrolysate (FPH) with pressure-driven tangential flow UF and following NF under high VRF value [164]. Solution of FPH1 (Prolastin, main MW 0.3−1 kDa) and FPH2 (MariPep C, main MW 1–4 kDa) were employed until reaching VRF of 6.0 and 8.3 in the NF step, respectively. They found out that UF was more impactful to fractionate less hydrolyzed FPH (FPH2) and resulted in higher permeate flux compared to NF, but high VRF cannot provide pure fraction of peptides as UF MWCO 4 kDa only produced permeate enriched with FPH below pivot points 0.6–0.75 kDa, and vice versa for retentate side [164]. The deficiency of UF-NF arrangement was also investigated during treatment of dairy effluent, even while shear-induced VSEP and RDM [165]. In part of NF, the absence of casein micelles that was removed beforehand in UF step accentuated the calcium ions to agglomerate the remaining proteins and decline the permeate flux at high VRF. Thus, neither improvement of permeate flux nor energy consumption can be picked up from this configuration. Further observation elucidates three configurations of: (1) UF-NF, (2) diafiltration-UF-NF, and (3) UF-diafiltration-NF for tuna dark muscle hydrolysates [166]. They concluded that configuration 1 was inefficient to fractionate this kind of protein hydrolysate, as NF retentate contained more than 80% of total peptide. Meanwhile, in configuration 2 and 3 NF retentate, the total protein was under 60%. Ultimately, although complete rejection of peptides above 4 kDa was performed by all variation, the permeate flux of configuration 3 was slightly lower owing to high protein content of UF permeate has triggered fouling. Additionally, insertion of diafiltration as a method to bridge the trade-off between yield and purity of the most bioactive peptides (NF permeate enriched with 0.3 kDa peptides) was also mentioned for fractioning peptides from white fish fillet [167].

Other approaches have been examined in order to optimize the selectivity, among others, membrane stacking of UF. Using RC membrane MWCO 30 kDa to separate myoglobin and β-lactoglobulin, 80.25%, 98.31%, and 100% of β-lactoglobulin rejection can be attained by one, two, and three membranes in parallel [168]. This research and other similar studies only examined model protein solution with binary or ternary compound [168–170]. Thus, the use of internally-staged dead-end UF for tilapia by-product hydrolysate (TBH) as the advancement from the previous research was one of the most representative case owing to the use of complex hydrolysate mixture [171]. Here, a single flat sheet RC UF membrane with MWCO 5 and 10 kDa, and multilayer orientation (top/bottom) of 10/5 and 5/5 kDa were compared in terms of permeate flux, selectivity, and ACE inhibitory activity. From the highest to the lowest permeate flux was developed by combination 10 > 10/5 > 5 > 5/5 kDa. This condition reflected that the first layer where the solution penetrates should be arranged in a way that makes concentration polarization is less intense, as in this case by positioning 10 kDa membrane, which has a more open structure (reducing concentration polarization), on top of the 5 kDa. Interestingly, permeate from 10/5 and 5/5 kDa multilayer combination suggested higher activities of 75.09 and 84.04%, respectively, as opposed to single membrane 5 kDa (71.83%) and 10 kDa (64.32%) since although all of them were enriched with peptides lower than 1.5 kDa, the two first were capable of specifically accumulating more peptides below 0.5 kDa [171].

Cascaded UFs with different stream configuration are sought to bring better performance. Patil et al. [172] applied three configurations of cascaded UF as displayed in Figure 5a–c to separate α-lactalbumin from β-lactoglobulin in a whey protein isolate. Under constant protein concentration in feed 2 g L$^{-1}$, pH 7.2, NaCl 5 mM, and cross-flow velocity 0.1 m s$^{-1}$, higher ratio of P1/W (product/waste) was integral to excellent separation performance. In one side, configuration C cannot manage the desired trade-off between yield and purity. Meanwhile, configuration A and B executed a more promising recovery and purity [172]. Following cascaded UF, other ways of UF-DF integration has been employed for several processes in protein separation. One of them was even elaborated more with EDBM [113], which is discussed in Section 2.3.

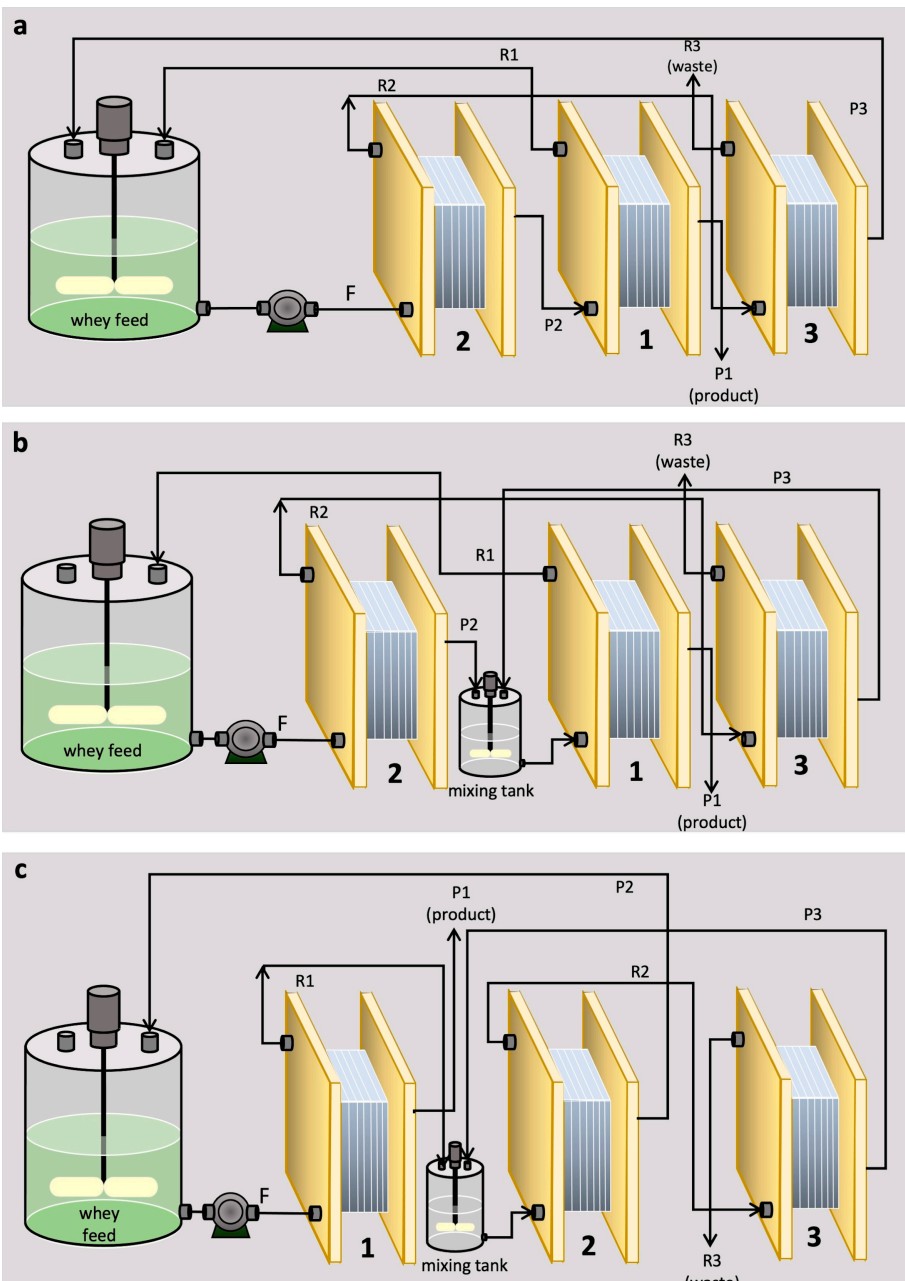

**Figure 5.** Cascaded UF with (**a**) non-constrained ideal; (**b**) adapted; and (**c**) countercurrent product recycle configuration. F: feed; P: permeate; R: retentate; W: waste.

Integration of jet-cooking with enzyme-assisted UF was revealed to increase the protein content of soy protein isolate (SPI) and nitrogen solubility index (NSI) and reduce isoflavones and phytic acid for infant formula [173]. UF membrane of 80 kDa was selected attributed to (1) increase of permeate flux from 4 to 12 L m$^{-2}$ h$^{-1}$, (2) slightly lower protein retention rate increase (<5% decline), (3) higher protein concentration in permeate (changed from 35 to 70 μg mL$^{-1}$), and (4) decline of phytic acid from 2.1 to 1.2 μg mL$^{-1}$ (all value compared between 30 kDa and 80 kDa, respectively). Jet-cooking temperature positively enhanced protein retention rate by aggregation of soluble proteins (higher MW), which was crucial for the efficiency of UF. Owing to energy optimization, 130 °C was considered to be the best.

## 3. Strategic Development of Ultrafiltration Membrane

### 3.1. Membrane Modules and Fiber Diameter

Different type of membrane module showed changes in efficiency and effectiveness to isolate specific proteins. As performed by the spiral and tubular polymeric modules that produced soy protein isolate, the later was capable of increasing the permeate flux (85 L m$^{-2}$ h$^{-1}$) at higher velocity rate (2.5 m s$^{-1}$) and 70 psi for 5% total solids. Meanwhile, the spiral modules only gave 19 L m$^{-2}$ h$^{-1}$ at 0.5 m s$^{-1}$ and 55 psi [174]. Moreover, spiral wound membrane is also limited to achieve higher VRF by high viscosity of retentate while compared to dynamic UF. However, combining this module with tubular ceramic membrane step would allow the VRF enhancement >8 (previously <5–6) [152]. Further consideration to opt the tubular polymeric modules is that they are endowed with lower operating and capital cost to reach the same performance. It should be emphasized that although different module was utilized, similar performance would be obtained if the same parameter β is used (detailed in Section 2.1). This result is reported during negatively charged UF of RC 100 kDa in a laboratory-scale flat-sheet and pilot-scale spiral-wound membrane (70× larger area) that both were available for retaining more than 98% whey protein [62].

Interestingly, membrane modules also determine specific amino acid residues retained or passed through the UF process. Of soy protein hydrolysate generated by consecutive pepsin and pancreatin treatment and following fractionation by hollow fiber and spiral-wound membrane with the same 10 kDa MWCO, there are four intriguing phenomena: (1) longer UF time of hollow fiber (3 h) led to higher content of glycine, threonine, valine, methionine, and isoleucine in permeate, representing smaller peptide as opposed to spiral-wound (10 min); (2) primary aromatic amino acids (tyrosine, leucine, and phenylalanine) produced by pancreatin (chymotrypsin) activity shows similar concentration in permeate of both membrane module; (3) amino acids constituting both low and high molecular-weight peptides, such as aspartic acid, serine, histidine, alanine, proline, arginine, and cysteine, have similar concentration in permeate and initial hydrolysate; and (4) highly retained amino acids (glutamic acid and lysine), which belong to high molecular-weight peptides, are more concentrated in the permeate of hollow fiber compared to spiral-wound. Yet, slightly higher antioxidative (ORAC assays) and antiviral activity (PFU assays) was found in the use of hollow fiber configuration [175].

Not only the type of module, but the diameter of the module influences the overall performance. Here, fiber diameter of a hollow fiber module was investigated to have impact on initial permeate flux and reversible fouling of pea protein concentrates [176]. Smaller lumen internal diameter has lower resistances and therefore must have better permeate flux according to Darcy's Law. It is exemplified that a 1.5 mm fiber module had $1.22 \times 10^{-5}$ m s$^{-1}$ initial flux, while that of 0.5 mm in diameter was $1.40 \times 10^{-12}$ m s$^{-1}$ for the same pressure and viscosity. The lower resistance is demonstrated to be due to minimum cake formation, induced by the predominant drag force that diminishes back-transport of foulants (this was not by means of shear rate).

### 3.2. Membrane Materials and Molecular Weight Cutoff

Various categories of membrane material were implemented in UF process. Inorganic membrane such as ceramic-derived is simpler in terms of construction, has less replacement cost, vapor-sterilizable, and able to achieve high CF [33,177,178]. Nevertheless, organic membranes, in particular, those that have hydrophilic features, are endowed by lower fouling tendencies. In fact, hydrophobic organic membranes like PES are still predominant in industrial use for their chemical resistances, robust mechanical strength, and thermal stabilities [178,179]. Although various laboratory works of membrane UF modification are well-established, those which used complex protein mixtures as testing solution are still scarce and therefore going to be showcased in this part.

One work researched the effect of material and molecular weight cutoff (MWCO) on the UF performances of saithe (*Pollachius virens*) protein hydrolysate using five tubular membranes (material variations: PES, modified PES (m-PES), and PS, and MWCO: 4, 6, 8, and 9 kDa) [180]. In that experiment, the objective was to separate peptides lower than 2 kDa that possess antioxidant activities. They concluded that considerable hydrophobic feature of PES created less permeable surface which is unfavorable for pertaining peptide separation. Permeate flux was more dependent on MWCO, and this value increased with the increase of MWCO. It is relevant to mention that depending on the hydrolysate the UF and membrane materials were working with, higher MWCO was not always directly correlated to the increase of permeate flux as membrane morphology (surface area, pore size and density) can drive different water permeability within the same membrane materials [181]. In addition, another work showed no connection between MWCO and yield-purity [182] since the homogeneity of membrane pore distribution is one of the keys to enhance selectivity [183].

Related to MWCO, because its actual value decreased from their nominal value due to the tested substances, operating conditions, and solute–solute–membrane interactions, researches to figure out the main effect of MWCO toward the UF performance needed to operate UF with low pressure and carefully choose membrane with MWCO above the smallest and below the biggest separated molecules [184]. The impact of MWCO on the characteristics of fouling is also relevant to the discussion. It is exemplified that while operated to fractionate aqueous extract of soy flour [185], lower MWCO of 50 kDa exhibited 0.2 μm foulant deposits in contrast to that of 100 kDa membrane with 0.4 μm thickness. Foulant was observed to have viscoelastic attributes, high resistance to shear stress in flat plate module, and low solid content (21.5% wt). The finding of thinner cake of smaller pore membrane is also coherent with the observation that for hydrolysate from corn ethanol process, smaller pore size (5 kDa) maintained slower fouling rate than 100 kDa membrane [182].

### 3.3. Surface Modifications

Surface modification to increase the performance is preferable in order to get the double advantages of structural support and hydrophilicity [183]. Next, this modification aims to increase flux (membrane performance), parallel to improve selectivity, as illustrated in Figure 6A,B, respectively. Low membrane permeability could be explored for additional modification to improve permeates flux owing to fouling reduction at the same time with resolving the selectivity issues.

One research aimed to functionalize charge properties of 100 kDa PES by cationic styrene polymerization inside the pore and following activation by sulfuric acid treatment to create open-structure PES [186]. Comparing to the pristine membrane, selectivity enhancement was nearly five-fold at pH 7.2 for β-lactoglobulin ascribable to lower membrane pore size (reduction in molecular sieving) and increase of electrostatic repulsion in negatively charged PES with β-lactoglobulin. Adjustment of pH would allow better selectivity as performed in a work that at pH 7.2, membrane was 50% more selective than at pH 3.2, of which this finding was supported by other researches [187,188].

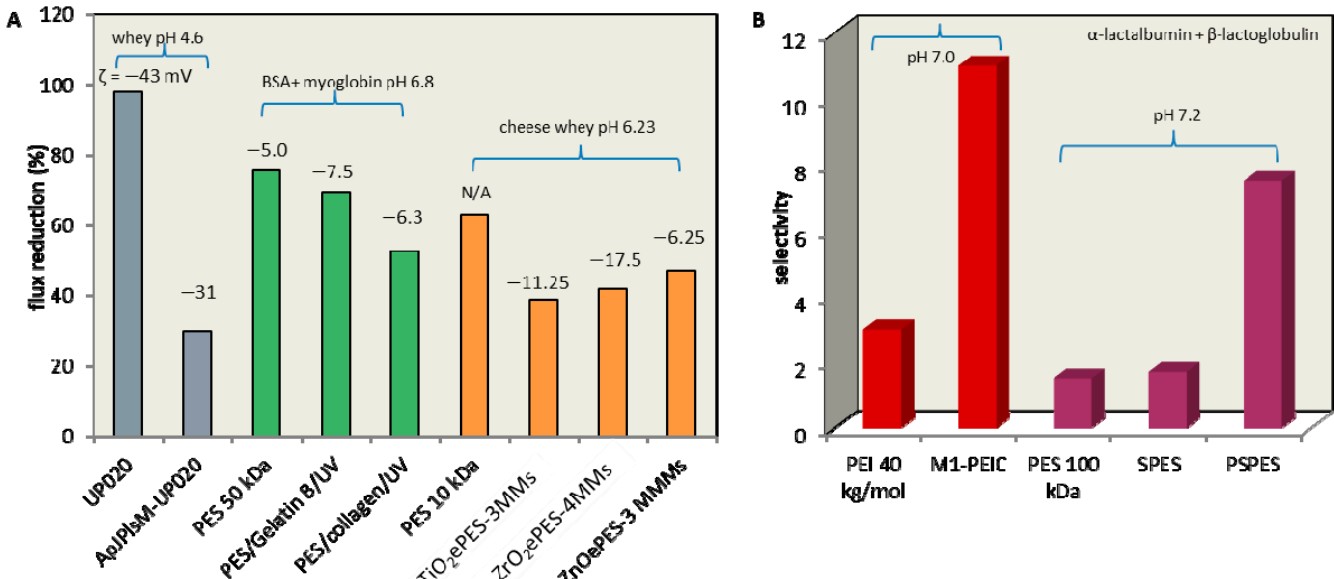

**Figure 6.** Performances of UF after selected surface modification (**A**) represented as flux reduction percentage and (**B**) selectivity. Data collected for (**A**) are from [189–191] and (**B**) from [186,188].

Regarding the hydrophilic enhancement, many strategies can be employed. *First*, this could be performed by metal-oxide nanoparticles such as $TiO_2$, $ZrO_2$ and ZnO to modify hydrophobic PES mixed matrix membrane by phase inversion technique [191]. When treating cheese whey effluent, 1.5 wt.% $TiO_2$, compared to 2.0 wt.% $ZrO_2$, 1.5 wt.% ZnO, and pristine membrane, also provided the highest average flux (25.09 L $m^{-2}$ $h^{-1}$) and lowest flux reduction (39%) owing to the increase in pore radius to be 4.16 nm and surface free energy (109.33 mJ $m^{-2}$). Following that, more than 94% of total whey protein could be rejected while individual protein rejection was 96% for BSA, and 90% for lysozyme and α-lactalbumin [191]. *Second*, not only via embedding metal oxide nanoparticles, atmospheric pressure jet plasma can alternatively elaborate hydrophilic feature on PES by incorporating oxygen-derived functional groups [183]. From this method, the ratio of reversible fouling to total fouling was decreased by 51% from 94% in pristine membrane, and formation of cake layer was wholly eliminated in parallel with reduced minimum roughness and transformation of maximum surface free energy from $-14.92 \pm 0.89$ mJ $m^{-2}$ to $+17.57 \pm 0.67$ mJ $m^{-2}$. Furthermore, when used to treat whey proteins, fouling rejection capability was equal with the initial condition. However, permeate flux promptly declined because of lessened hydraulic permeability emerged from plasma-induced surface cross-linking. Nonetheless, permeate flux recovery after cleaning with 0.1 N NaOH was obtained in a great result while using this modified membrane [183].

## 4. Ultrafiltration Performance and Selectivity Enhancements: Controls of Fouling and Concentration Polarization from Operating Conditions Perspectives

Fouling and concentration polarization influence the dynamics layer of membrane surface, which in turn control the mass transfer during UF process [192]. Generally, fouling built-up ranges over the duration of UF process, whereas concentration polarization is only below one minute and reversible in nature [29,193]. In this paper, we gathered distinguished examples of how operating parameters play a role to control fouling and concentration polarization, i.e., protein concentration, pH, and ionic strength (salt concentration) by modulating protein–protein and protein–surface interactions, and TMP toward hydrodynamics. Owing to the unique findings resulted from different types of protein sources, the discussion of each operating parameters will be divided into marine, dairy and legume proteins. In Section 4.1, membrane characteristic like hydrophilicity, MWCO and UF mode were taken into discussion. By and large, there is a trade-off between all of

these parameters to generate efficient and effective UF process, which again unique for each kind of protein being employed.

### 4.1. Protein Concentration

High concentration of feeds (peptides) allows the development of severe concentration polarization and fouling. Subsequently, this can lead to either increase or decrease of retention factor (RF). First, RF increase will be due to the compact and less permeable layer is formed which in turn increasing filtering capacity and decreasing permeate flux Figure 7. This can be also attributed to the lower MWCO of the membrane. On the contrary, the decrease of RF can be as consequence of viscous polarization layer that prevents counteraction of solute accumulation by retro-diffusion [194,195]. It is interesting to note that in binary model solution, both mass flow and RF of a compound will follow the trends of other peptides which are highly concentrated. For example, myoglobin of tuna concentrate was 10-fold more than its hemoglobin content; thus the mass flow of hemoglobin was higher at pH 8.6 (isoelectric point (IEP) of myoglobin), not at pH 7.3 (IEP of hemoglobin) [196].

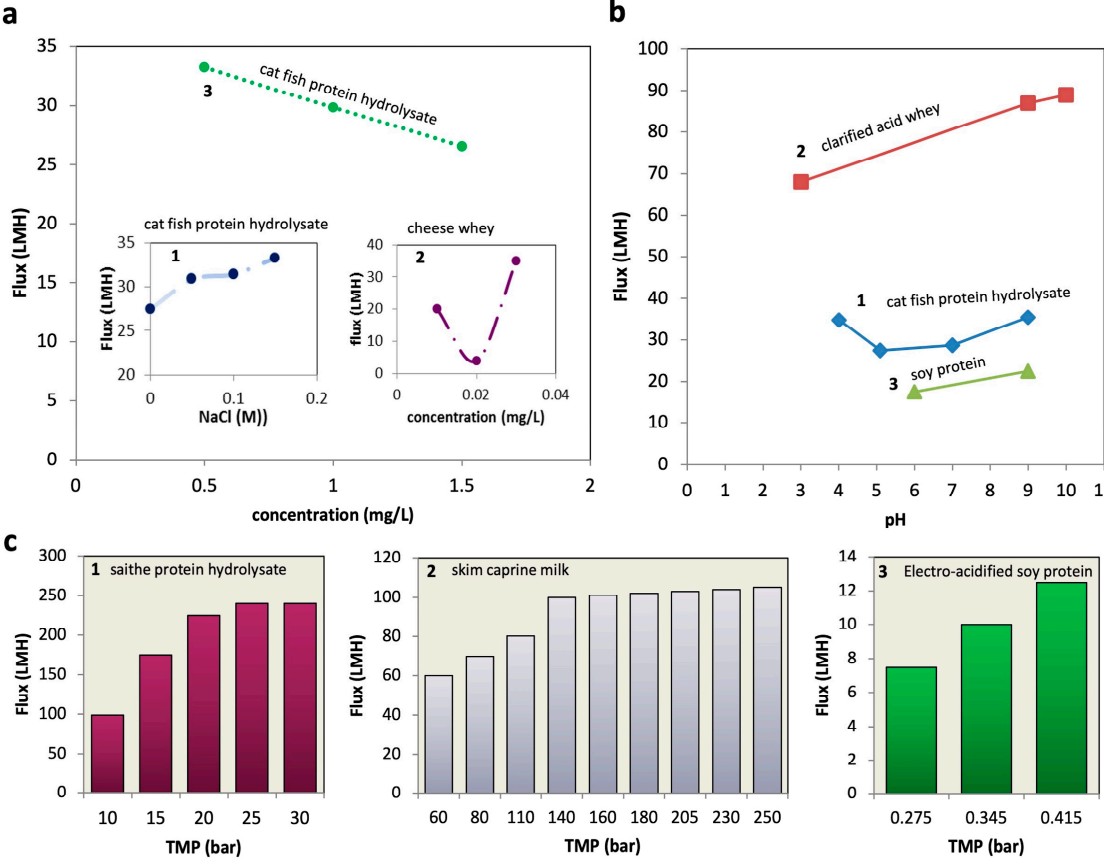

**Figure 7.** The effects of various parameters on UF flux (**a**) protein concentration and ionic strength; (**b**) pH; (**c**) TMP. Membranes in a1, a3: RC 5 kDa [197], a2: PS 30 kDa [198], b1: RC 5 kDa [197], b2: ceramic membrane $ZrO_2$-$TiO_2$ (0.045 m$^2$) [177], b3: PS hollow fiber 100 kDa (this data) after EDBM with Electrocell AB (eff electrode area 100 cm$^2$) [199], c1: tubular PES 4 kDa [194], c2: tubular ceramic 4 kDa [200], c3: hollow fiber PS 100 kDa [201].

The pattern of RF reduction was sometimes not linear along with the increase of protein concentration. As being shown here, whey milk concentrate (WPC) 80% processed by using 30-kDa PS membrane suggested a unique manner of protein concentration toward flux. Permeate flux increased as following 2%-wt < 1%-wt < 3%-wt protein solution since at some point, the deposition rate may lower than the removal rate like in 3%-wt protein, and thus increasing concentration does not merely trigger up the fouling formation [198]

(Figure 7a). In some researches, denatured proteins could also affect the performance of UF. One research analyzed UHT milk which contains denatured whey protein more than the pasteurized milk, but with the same native whey protein concentration. There, UF of pasteurized milk using 50 kDa ceramic membrane in a rotating disk module led to the higher permeate flux. This trend was supported by another work that used M5 Carbosep membrane 10 kDa to fractionate UHT skim milk and low-heat skim milk powder [202].

Feed concentration also affects the migration rate of protein. For instance, in EDUF system, linear increase of peptide concentration in feed leads to higher migration rate as performed in peptide solution containing Asp, Glu, His, and Arg that reached maximum of 16.2 and 7.8 g m$^{-2}$ h$^{-1}$ at 4% feed concentration for cationic and anionic compartments, respectively. In this work, energy consumption also declined from 17.4 to 3.53 W h g$^{-1}$ while peptide concentration climbed up from 0.5% to 4% [203].

*4.2. pH in Pretreatment*

Various research studies provided information on the effect of pH on UF performance. Severe fouling can occur in operation at IEP, leading to lower permeate flux [30,157,204,205]. However, it is also expected that at IEP, the transmission of protein become the highest due to the minimum electrostatic repulsion (charged effect) [157]. Interestingly, because of the denser fouling layer developed by employing smaller protein size, the permeability was the lowest at IEP [206].

Intermolecular electrostatic repulsion could be achieved by arranging operating pH higher or lower than majority's IEP, so there will be an added sum of negative- or positive charged molecules that inhibits fouling. However, this condition does encumber the target of permeate flux. While limited peptides exist in the medium, one can easily arrange the process at IEP of targeted compound, demonstrated here by fractioning myoglobin from hemoglobin of yellowfin tuna red muscle. They implemented pH 8.6 which was the IEP of myoglobin and therefore, the charged hemoglobin (IEP 7.3) will have positive charge and minimal mass flow in contrast to myoglobin. Furthermore, in pH 8.6, with higher TMP, the mass flow tended to increase (myoglobin, 12 g m$^{-2}$ min$^{-1}$ and hemoglobin, 0.8 g m$^{-2}$ min$^{-1}$). As the net charge of myoglobin is zero, its molecule expressed the lowest bulk mass transfer diffusion and thereby concentration polarization can contribute in a positive way that increase protein transmission [196].

The same procedure of tuning the pH also takes place while it comes to complex protein solution even though unlike experiment using binary solution model, permeate flux might be lower and its control is harder due to intervention of other factors such as concentration, TMP, etc. Represented during CFPH fractionation, after 20 min at pH 4.0 and 9.0, permeate flux was 37.63 and 39.20 L m$^{-2}$ h$^{-1}$, respectively, but in their IEP (pH = 5.1), it was only 35.54 L m$^{-2}$ h$^{-1}$. Additionally, the flux reduction while operation in IEP was nearly 10 L m$^{-2}$ h$^{-1}$ as opposed to in average 5 L m$^{-2}$ h$^{-1}$ for pH 4.0 and 9.0 [197]. Plausible reason for this was that in highly acidic or basic environment, high electrostatic repulsion hindered the peptide transmission (40% in pH 4.0, 7.0, and 9.0) which did in fact reach maximum at IEP (48.81%) [197] (Figure 7b). After all, as transmission of desired peptide is far more important that permeate flux of all proteins, operation is usually selected in their IEP point.

With respect to plant-based protein, antinutrients like phytic acid and trypsin inhibitors are aimed to be separated from the rest. Removals of phytic acid in soy protein isolate using diafiltration technique showed that pH of 9.0 was not favorable compared to 6.5 [207]. This was because the higher positive charge of phytic acid could stimulate ternary complex of cation-phytic acid, which made removal efficiency to decline. Interestingly, in pH 6.0 of tangential UF of protein concentrates from chickpea with 50 kDa hollow fiber membrane, UF pH 9.0 followed by DF pH 6.0 was capable of reducing the phytic acid to the lowest level that cannot be achieved by isoelectric precipitation at pH 4.5 [208]. Meanwhile trypsin inhibitor content also did not see any changes with all employed variation of single stage UF. So, the reduction of phenolic compounds was better held in the system

of UF pH 9.0 and DF pH 9.0. Furthermore, in another finding for producing low-phytate soy protein isolate by EDBM, reducing pH of UF operation from 9.0 to 6.0 will not only accelerate the rate of acidification by two times, but also lower fouling tendencies (total cell resistances) of protein deposition in ED spacers which give 20%-increase on permeate flux [199].

In EDUF, pH did not affect the migration rate along operation, but determined peptide composition inside the compartments, like illustrated by pepsin-pancreatin soy protein isolate which used EUR-2C cell with effective area 200 cm$^2$ (6 UF membrane 10 kDa) [209]. The conductivity of KCl$_1$ space (anionic fraction) at pH 3.0 did not show downward trend as other variation at pH 6.0 and 9.0 in KCl$_1$ and pH 3.0, 6.0 and 9.0 in KCl$_2$ (cationic fraction) due to higher weight peptide in feed. Furthermore, at pH 9.0, the selectivity of cationic peptides lower than 0.4 kDa in KCl$_2$ increased since global peptide charge was negative (IEP equal to 4.5) and with negative surface of PES, the repulsion was the maximum in connection with diminishing anionic diversity in KCl$_1$.

### 4.3. Ionic Strength

In UF, pH is primarily attributed to the altercation of fouling rate and ionic strength is on protein–surface or protein–protein interactions [157]. When many approaches target small peptides with higher transmission rate at operation in its IEP, adding NaCl can improve both the permeate flux and transmission. This value in one instance corresponded to 27.49 L m$^{-2}$ h$^{-1}$ and 48.81% transmission (no salt) to 33.29 L m$^{-2}$ h$^{-1}$ and 52.95% transmission (0.15 M NaCl) of CFPH fractionation. Higher ionic strength directed the anion binding to peptide, creating bigger size peptide and facilitating the transport of small peptides to retentate [197]. Nevertheless, other mechanism may take in charge of influencing peptide transmission. To illustrate, in fractioning myoglobin from hemoglobin, RF reached its peak while at higher concentration of NaCl because of self-dissociation of myoglobin doubled its apparent molecular weight [196].

Ionic strength gives a positive impact in the separation of protein in EDUF. It was found that the increment in ionic strength accentuates the negative charge density of UF membrane, directing a more effective pore size by virtue of electrostatic repulsion between ions and thinner hydration layer caused by salting out effect at pore walls [210]. Moreover, 1, 3, and 5 g L$^{-1}$ KCl in recovery compartment amplified the relative abundance of arginine and lysine of snow crab by-product hydrolysate and overall peptide migration rate (13.76 $\pm$ 3.64 g m$^{-2}$ h$^{-1}$) [210].

In prior works with various UF membranes, high shear rate tangential UF [110], sequential UF/DF [113], and electroacidification-UF [80] potentially managed phytic acid reduction, but then encountered the fouling problem. Pretreatment by KCl extraction alleviated the phosphorus to protein content (representing phytic acid content) during UF and diafiltration (DF) of soy protein isolate. Furthermore, although extraction procedure using water and KCl gave a similar fouling resistance, implementation of backwashing was not compulsory for KCl-mediated process on account of lower protein contents which may create compact fouling layer in UF step [207]. More importantly, the highest protein fraction they got was 91.8% with 0.06 M KCl, pH 9.0, 25 °C (extraction), pH 9.0 (UF), and pH 6.5 (DF). The result for that study was also confirmed in other works in soy protein isolate [199,211] and in pea protein isolate [176,208]. Despite all those advantages, a study that addressed the effect of pH on particle size distribution presented the condition of adding 0.12 M KCl to produce soy protein extract will result in the decrease of permeate flux with lower pH. For instance, in pH 6.0, the protein aggregates are possibly to grow due to minimum electrostatic repulsion, particle volume fraction, and protein transport to the concentrated layer was facilitated [205,211,212].

### 4.4. Operating Pressure

When it comes to TMP, higher value corresponds to increase in flux. However, to some extent, the permeation flux is not proportional toward the increase of TMP according

to Darcy's Law. Despite during the fractionation of saithe protein hydrolysate with Micro-lab40 pilot plant UF/NF membrane (recirculation mode), no concentration polarization was observed under 15 bar, reversible polarization that leads to nonlinear flux emerged between 15 and 25 bar. In the next step, further pressure increase made the flux becoming independent variable from TMP [194] (Figure 7c). Adjusting TMP also give an impact on selectivity, along with initial peptide concentration and VRF. With recirculation mode, under 30 bar and 30 g L$^{-1}$ saithe protein, most of the peptides lower than 1 kDa can be collected in the permeate side [194]. Yet, TMP determine the sieving characteristics of UF membrane since MWCO decreased (retention rate higher) while TMP increased because of the elastic deformation of membrane pores [194]. However, complex hydrolysate mixtures resulted in opposite outcome due to low fouling properties of dextran solution [213].

In order to increase permeate flux of electroacidified soy protein by cross-flow UF, a computational fluid dynamics study showed that increasing TMP determined more significantly as opposed to enhancing axial velocity which only provide mediocre permeate flux increase [201] (Figure 7c). Regarding the fouling, it is worth to note that firstly, irreversible fouling had to be manipulated not by tuning the TMP, but by the viscosity of hydrolysate (concentration of protein and inherent minerals) in a complex mixture since this parameter is the most sensitive amongst all. Secondly, diffusion coefficient is undoubtedly a major influence of the protein back-transport from membrane surface for nonelectroacidified soy protein hydrolysate [201].

Different targets of final products require unlikely the TMP optimization. In producing high content of oligosaccharides from goat milk by cross-flow diafiltration up to 4 DV with 50 kDa tubular ceramic membrane, higher TMP transmitted more protein (11%) into the permeate side, which even though favorable to decrease operation time by 10.5 h, lowering the oligosaccharides fractions [200]. While treating feta cheese whey to develop powder with high content of lactoferrin and IgG by 100 kDa cylindrical polyvinylidene fluoride (PVDF) membrane, high TMP (5 bar) is attributed to fouling and concentration polarization, despite to some point increase the permeate flux which has no correlation with better targeted compounds [214]. In contrast, in order to separate specific peptides from whey, one case showed that transmission of α-lactoglobulin, derived from low-heat skim milk powder, declined by 0.13 point from 0.78 in the increase of TMP between 40 and 80 kPa. However, β-lactoglobulin transmission was improved to be 0.35 (initially 0.30) in the same changes of TMP [148].

*4.5. Temperature*

Temperature of UF process would influence not only in the enhancement of permeate-side flux, but also mass transfer and kinetics of protein denaturation. Optimization of temperature for plant protein separation is still limited. An example of clarification of depectinized kiwifruit juice by ultrafiltration suggested that flux increased with temperatures from 20 to 30 °C and with axial feed flow rate from 300 to 700 L h$^{-1}$. From these conditions, cake layer and irreversible fouling resistances contributes as much as 2.23% and 2.75%, respectively, to the total resistance, whereas the contribution of the reversible fouling was more pronounced (29.4%) [215]. Increase of the flux was also found in milk concentration and fractionation by tubular membrane to attain total solids 21.55% and 8.6% protein, the flux was higher by 16% at 50 °C, compared to 45 °C, and this increase will be more powerful at faster fluid velocity. Generally, this impact of temperature follows the Arrhenius correlation, with average activation energy of 6.8 ± 8 kcal mol$^{-1}$. However, they demonstrated that rejection coefficients of proteins, fat, and lactose are not a function of temperature, pressure, and fluid velocity [216]. So, levelling the temperature up to 70 °C will still increase the permeate flux owing to the facilitation of mass transport by lower viscosity [175].

With respect to increase in CF, different optimum temperature may regulate as exhibited by production of milk protein concentrates (MPC) with five-fold concentration using PES membrane under 15, 30 or 50 °C [217]. They investigated a lower size of casein

micelles at 50 °C (83.6 nm) from 15 and 30 °C (92.7 nm), reflecting furthermore the changes in protein conformation owing to the denaturation started at 50 °C. This resulted in severe fouling (when reacted with calcium ions) and reduction in permeate flux even at higher temperature. In addition, processing at temperature 15 °C was finally showed to have exceeding emulsifying characteristics (mineral balance) without afflicting heat stability and solubility [217]. Moreover, sensory characteristics of dairy products, e.g., feta whey protein concentrates, were deeply altered by higher temperature (20 °C was the optimum value), such as turning the color of retentate and final powder (after freeze drying) from grey to green. The increase of temperature from 30 °C to 40 °C enabled higher immunoglobulin yield, but lower their temperature-sensitive lactoferrin [214].

## 5. Conclusions

The development of UF for protein concentration and selective separation encounters more often than not fouling and concentration polarization phenomena. The resurgence of concentration polarization is initialized during the first minute of UF and has reversible feature, in contrast with fouling that is induced over the process duration. These two challenges influence the performance in an individual manner, depending on protein being employed, operating conditions, and membrane configurations. Overall, they generate the trade-off between productivity and selectivity of the process. In order to tackle these challenges, some advancements are developed in protein ultrafiltration.

Some reported studies show the advantages of using charged membrane during protein separation, such as higher separation rate, flexible design, and simpler system than pressurized UF. However, the separation is only effective for proteins that have far different IEP, low electrolyte conductivity, and productivity (only in the level of milligrams per hour). Combining UF with ED is useful for separating protein with similar molecular weight but with different charge. Amidst all of the advantages in operating EDUF, water splitting, fouling, and vulnerable membrane integrity still obstruct the scale-up strategies. EDUF could ensure minor degradation of UFM, but the protein fouling on ion-exchange membrane cannot be ignored.

Fouling is the major problem of membrane operation, including in ultrafiltration of protein. Numerous studies have been devoted to find a method for controlling fouling formation, such as ultrasonic-assisted UF, dynamic UF, and HPTFF. These methods can suppress fouling phenomena by breaking up cake layer on the membrane surface and decreasing concentration polarization. Another way for controlling fouling formation is by tuning filtration conditions, such as concentration, pH, ionic strength, transmembrane-pressure, and temperature. Optimum adjustment of operating condition can result in both high separation performance and fouling reduction.

Membrane modification is also an interesting way for performance improvement of UF. Surface modification aims to get the double advantages of structural support and hydrophilicity. Interestingly, low membrane permeability could be tailored for more to improve permeates flux owing to fouling reduction at the same time with solving the selectivity needs. Until now, not so many of membrane modification have been tested by complex protein hydrolysate mixtures, and thus open up the chances of further research. Yet, the effect of electrostatic interactions on the observed protein sieving could be more elaborated. Surface modification may be directed to introduce positive or negative charges on the membrane surface providing electrostatic repulsion. The repulsion mechanism may help to reduce the deposition of foulant on the membrane surface.

As the UF in commercial process is joined together with other separation units, each performance, including energy requirements, purity, yields, bioactivities, and economical flux, needs to be evaluated as a whole. In dairy industries, for example, the deficiency of UF-NF arrangement was investigated during treatment of dairy effluent. Advantageously, diafiltration can be incorporated to achieve the desired flux. Other techniques such as integration of UF and MF, cascaded UF, UF membrane stacking, and jet-cooking

with enzyme-assisted UF was revealed to increase the protein content while delivering reasonable flux.

**Author Contributions:** Writing—original draft preparation, E.R., R.R. and K.K.; writing—review and editing, R.B., and I.G.W.; conceptualization, I.G.W.; funding acquisition, I.G.W. All authors have read and agreed to the published version of the manuscript.

**Funding:** This research is funded by Indonesian Ministry of Research and Technology/National Agency for Research and Innovation, and Indonesian Ministry of Education and Culture under WCU Program managed by Institut Teknologi Bandung.

**Institutional Review Board Statement:** Not applicable.

**Informed Consent Statement:** Not applicable.

**Data Availability Statement:** No new data were created or analyzed in this study. Data sharing is not applicable to this article.

**Conflicts of Interest:** The authors declare no conflict of interest.

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
