# Peer review of "Recent Advancements of UF-Based Separation for Selective Enrichment of Proteins and Bioactive Peptides—A Review"

_applsci, doi:10.3390/app11031078_

Round 1
Reviewer 1 Report
Protein and peptide concentration, purification is important as these are inevitable nutrients. Ultrafiltration is one of the major protein concentration methods, which has been studied for long mainly in food industry. This review paper focuses on recent advancement in protein concentrating method associated with ultrafiltration including newly developed technologies.
Authors reviewed significant amount of research papers related with protein concentration by ultrafiltration and relevant new technologies.
These huge number of reviewed research itself is really informative, and those effort could be highly appreciated.
However, the structure of the article should be improved. And also, interpretations and explanation of reviewed research should be improved to be meaningful review.
Although the author's focus is on the review of the latest research about protein concentration by ultrafiltration based methodologies, basic information and key issues should be described first before the latest research review.
Otherwise, the significance of the latest research cannot be interpreted properly due to the lack of comparative information.
As for the review of the latest research, author listed significant amount of research on the advancement and newly developed technologies around protein concentration associated with ultrafiltration. However, the most of contents for those parts (section 2) are rather just the list of results of reviewed articles. And few useful interpretation and scientific explanations as underlying mechanisms are described. For example, introduction of newly developed technologies requires underlying scientific mechanisms even if these mechanisms are not fully understood. Thus, the review of the latest research, actually this seems to be the main part of this review paper, is not that much meaningful.
From these view points, I recommend that a major revision is warranted, even though the huge amount of reviewed research is informative. I explain my concerns in more detail below. I ask that the authors specifically address each of my comments in their response.
===== Specific comments
>Abstract
The contents of abstract is not the abstract of this review, rather like introduction.
> p.2 L 65, L68
The terms "membrane UF", "protein UF" and "peptide UF" are not appropriate terminology.
> Section 2
Even though the focus of this section is on the advancement in ultrafiltration for protein separation, underlying mechanisms for methodologies in those latest research should be explained. Otherwise, these description is merely the list of results of reviewed research. Additional to this, the explanation of each technology is not clearly understandable because the too many results are listed and the important information may get lost.
> Section 2.1
If the charge of membrane is referred, the material of membrane should be described here also, because the membrane charge is chemically determined by functional groups of composing material.
> Figure1 and Figure2
There is no referring of Figure 1, and Figure 2c~2f in the manuscript. The explanation is required for those figures.
Also, it is not clear that those schematics are original or from reviewed articles.
> For section 2.3
There is no explanation about "bipolar membrane" for EDBM technology.
> For section 4
Most of contents in the section 4 are basics and widely understood already for membrane filtration including UF. Focus of this section is not clear, even though the many articles are referred.
> Overall
The organization of this manscript could be improved to be more informative and useful review.
Author Response
Reviewer # 1
Protein and peptide concentration, purification is important as these are inevitable nutrients. Ultrafiltration is one of the major protein concentration methods, which has been studied for long mainly in food industry. This review paper focuses on recent advancement in protein concentrating method associated with ultrafiltration including newly developed technologies. Authors reviewed significant amount of research papers related with protein concentration by ultrafiltration and relevant new technologies. These huge number of reviewed research itself is really informative, and those effort could be highly appreciated.
However, the structure of the article should be improved. And also, interpretations and explanation of reviewed research should be improved to be meaningful review. Although the author's focus is on the review of the latest research about protein concentration by ultrafiltration based methodologies, basic information and key issues should be described first before the latest research review. Otherwise, the significance of the latest research cannot be interpreted properly due to the lack of comparative information. As for the review of the latest research, author listed significant amount of research on the advancement and newly developed technologies around protein concentration associated with ultrafiltration. However, the most of contents for those parts (section 2) are rather just the list of results of reviewed articles. And few useful interpretation and scientific explanations as underlying mechanisms are described. For example, introduction of newly developed technologies requires underlying scientific mechanisms even if these mechanisms are not fully understood. Thus, the review of the latest research, actually this seems to be the main part of this review paper, is not that much meaningful. From these viewpoints, I recommend that a major revision is warranted, even though the huge amount of reviewed research is informative. I explain my concerns in more detail below. I ask that the authors specifically address each of my comments in their response.
Specific comments
- Abstract: The contents of abstract is not the abstract of this review, rather like introduction.
Reply: We thank the reviewer for his/her helpful comment. We have revised the text to
“Proteins are one of the primary building blocks which have significant functional properties to be applied in food and pharmaceutical industries. Proteins could be beneficial in their concentrated products or isolates, of which membrane-based filtration like ultrafiltration (UF) encompasses application in broad spectra of protein sources. More importantly, selective enrichment by UF is of immense interests due to the presence of antinutrients that may dominate their perspicuous bioactivities. UF process is obstructed by primarily concentration polarization and fouling that in turn emerge as a trade-off between productivity and selectivity, especially when pure isolates are an ultimate goal. Number of factors such as operating conditions and membrane equipment could leverage those pervasive contributions, and therefore UF protocols should be optimized for each unique protein mixture and mode of configuration. For instance, employing charged UF membranes or combining UF membrane with electrodialysis enable efficient separation of proteins with a similar molecular weight that is hard to achieve by the conventional UF membrane. Meanwhile, fouling control strategies such as by utilizing ultrasonic waves, tuning operating conditions (e.g. pressure, pH, and ionic strength), and modifying membrane surface can mitigate fouling effectively and can suppress the concentration polarization phenomena. A plethora of advancements in UF, from their membrane material modification to the arrangement of new configurations are a quest to actualize promising potentials of protein separation by UF and it is reviewed in this paper.”
- 2 L 65, L68: The terms “membrane UF”, “protein UF” and “peptide UF” are not appropriate terminology.
Reply: We thank the reviewer for his/her helpful comment. We have revised the text to
“The advent of those novel advancements in UF is owing to the bigger challenges in managing the trade-off between productivity and selectivity.”
“It is obvious that published researches in UF for peptide separation (see the inset) are always lower than UF for protein separation due to the setbacks of UF to achieve high selectivity of small peptide fractions regardless of their higher productivity.”
- Section 2: Even though the focus of this section is on the advancement in ultrafiltration for protein separation, underlying mechanisms for methodologies in those latest research should be explained. Otherwise, these description is merely the list of results of reviewed research. Additional to this, the explanation of each technology is not clearly understandable because the too many results are listed and the important information may get lost.
Reply: We thank the reviewer for his/her helpful comment. We have improved the discussion in section 2 separation of protein by charged membrane and EDUF.
- Section 2.1: If the charge of membrane is referred, the material of membrane should be described here also, because the membrane charge is chemically determined by functional groups of composing material.
Reply: We thank the reviewer for his/her helpful comment. The charged UF membrane have been mentioned in Table 4.
- Figure 1 and Figure 2: There is no referring of Figure 1, and Figure 2c-2f in the manuscript. The explanation is required for those figures. Also, it is not clear that those schematics are original or from reviewed articles.
Reply: We thank the reviewer for his/her helpful comment. Figure 1a and 1b have been cited in introduction, while figure 2a and 2b have been cited in sub-section 2.2. There are no figure 2c-2f.
- For section 2.3: There is no explanation about “bipolar membrane” for EDBM technology.
Reply: We thank the reviewer for his/her helpful comment. We have added new explanation about bipolar membrane in section 2.3.
- For section 4: Most of contents in the section 4 are basics and widely understood already for membrane filtration including UF. Focus of this section is not clear, even though the many articles are referred.
Reply: We thank the reviewer for his/her helpful comment. In section 4, we gathered distinct examples of how operating parameters play a role to control fouling and concentration polarization, i.e.: protein concentration, pH, and ionic strength (salt concentration) by modulating protein-protein and protein-surface interactions, and TMP towards hydrodynamics. This section also discusses the effect or each parameters on flux during protein ultrafiltration. We believe that the gathered information from recent studies would give insights for operating UF in protein separations.
- Overall: The organization of this manuscript could be improved to be more informative and useful review.
Reply: We thank the reviewer for his/her helpful comment. We have added some explanation in section 2 to make the manuscript more informative.
Reviewer 2 Report
It makes me difficult to judge a work done of compiling publications already peer-reviewed.
It seems to me that authors do an exhaustive bibliographic study of the subject, but remaining in the qualitative treatment. In my opinion, a review should also include a basic theoretical part of each of the processes discussed. That is the reason because I consider that this work may be improved.
From the editing point of view, I think that the resolution of the figures must be improved (for example, figure 3) or the units must be properly written in a correct format. Authors have to improve, for example, the units of lines 174, 175, 193, 216 ... and write them properly as it has been well done on line 359.
Author Response
It makes me difficult to judge a work done of compiling publications already peer-reviewed. It seems to me that authors do an exhaustive bibliographic study of the subject, but remaining in the qualitative treatment.
- In my opinion, a review should also include a basic theoretical part of each of the processes discussed. That is the reason because I consider that this work may be improved.
Reply:
- From the editing point of view, I think that the resolution of the figures must be improved (for example, figure 3) or the units must be properly written in a correct format.
Reply: We thank the reviewer for his/her helpful comment. We have improved the quality of the figures.
- Authors have to improve, for example, the units of lines 174, 175, 193, 216 ... and write them properly as it has been well done on line 359.
Reply: We thank the reviewer for his/her helpful comment. We have written the units properly. Revisions are highlighted in the revised manuscript.
Reviewer 3 Report
This MS reviews the most recent advancements in ultrafiltration techniques for the isolation of peptides and proteins. The work is complete and well organized. It certainly deserves publication.
Some minor revision:
- l. 34: proteins are essential
- l. 42: proteins
- l. 57: please revise this sentence and check English
- l. 170: selectivity
Author Response
This MS reviews the most recent advancements in ultrafiltration techniques for the isolation of peptides and proteins. The work is complete and well organized. It certainly deserves publication.
Some minor revision:
- 34: proteins are essential
Reply: We thank the reviewer for his/her helpful comment. We have revised the text to
“Proteins are essential to maintain proper energy density of living beings, along with regulating activities of enzymes pertinent to type-2 diabetes, hypertension, and stress relief [1]”
- 42: proteins
Reply: We thank the reviewer for his/her helpful comment. We have revised the text to
“Proteins are essential to maintain proper energy density of living beings, along with regulating activities of enzymes pertinent to type-2 diabetes, hypertension, and stress relief … ”
- 57: please revise this sentence and check English
Reply:
Reply: We thank the reviewer for his/her helpful comment. We have revised the text to
“Therefore, multitudes of experimental work, which many have been integrated into commercial processes, are initiated to advance UF with a combined driving force (electrical and/or concentration).”
- 170: selectivity
Reply: We thank the reviewer for his/her helpful comment. We have revised the text to
“It was said that the selectivity was enhanced to be above 800 owing to the electrophoretic effects that allowed the filtration velocity to be kept high at prolonged time”
Round 2
Reviewer 1 Report
The manuscript has been improved, and this manuscript can be acceptable after correction of Figure 4 has been done.
Author Response
The manuscript has been improved, and this manuscript can be acceptable after correction of Figure 4 has been done.
Reply: We thank the reviewer for his/her helpful comment. Figure 4 was corrected
Reviewer 2 Report
Thank your very much to the authors to improve the manuscrip. However it must be revised once more, becuase authors have been confused when numbering the figures. There are 2 "figure 1" (page 3 and page 13) and from there all the numbering is wrong.
In the revised version, the figure on page 22 is empty. I don't know if it is pdf conversion problem.
They must check also the decimal separation. I have detected some errors, for example in line 761 "pH 4,0".
Author Response
Thank you very much to the authors to improve the manuscript. However, it must be revised once more, because authors have been confused when numbering the figures. There are 2 "figure 1" (page 3 and page 13) and from there all the numbering is wrong.
Reply: We thank the reviewer for his/her helpful comment. We have revised the numbering. It could be error during the pdf conversion.
In the revised version, the figure on page 22 is empty. I don't know if it is pdf conversion problem.
Reply: We thank the reviewer for his/her helpful comment. It could be error during the conversion.
They must check also the decimal separation. I have detected some errors, for example in line 761 "pH 4,0".
Reply: We thank the reviewer for his/her helpful comment. We have checked the decimals.